


# A Parallel Hybrid Intelligence Algorithm for Solving Conditional Nonlinear Optimal Perturbation to Identify Optimal Precursors of North Atlantic Oscillation

Bin Mu[1], Jing Li[1], Shijin Yuan[1], Xiaodan Luo[1], and Guokun Dai[2]

[1]Department of Software Engineering, Tongji University, Shanghai, China
[2]Department of Atmospheric and Oceanic Sciences & Institute of Atmospheric Sciences, Fudan University, Shanghai, China

**Correspondence:** Shijin Yuan (yuanshijin2003@163.com)

**Abstract.** The North Atlantic Oscillation (NAO) is the most prominent atmospheric seesaw phenomenon in North Atlantic Ocean. It has a profound influence on the strength of westerly winds as well as the storm tracks in North Atlantic, thus affecting winter climate in Northern Hemisphere. Therefore, it is necessary to investigate the mechanism related with the NAO events. In this paper, conditional nonlinear optimal perturbation (CNOP), which has been widely used in research on the optimal
precursor (OPR) of climatic event, is adopted to investigate which kind of initial perturbation is most likely to trigger the NAO anomaly pattern with the Community Earth System Model (CESM). Since CESM does not have an adjoint model, we propose an adjoint-free parallel principal component analysis (PCA) based genetic algorithm (GA) and particle swarm optimization (PSO) hybrid algorithm (PGAPSO) to solve CNOP in such a high dimensional numerical model. The results demonstrate that the OPRs obtained by CNOP trigger the reference flow into typical NAO mode, which provide the theoretical underpinning
in observation and prediction. Furthermore, the hybrid algorithm can accelerate convergence and avoid falling into a local optimum. After parallelization with Message Passing Interface (MPI) and Compute Unified Device Architecture (CUDA), the PGAPSO algorithm achieves a speed-up of $40\times$ compared with its serial version. The results as mentioned above indicate that the proposed algorithm can efficiently and effectively acquire CNOP and can also be generalized to other complex numerical models.

# 1 Introduction

The North Atlantic Oscillation (NAO) refers to the continuous phase-reversing oscillation in the meridional direction of the sea level pressure (SLP) field in the North Atlantic, which is mainly related to the interannual variation of the pressure over Azores and Iceland. It represents large-scale alterations in the SLP differences between the subtropical and subpolar regions of the North Atlantic (Haylock et al., 2007). As the dominant mode of atmospheric circulation variability in the northern hemisphere,
the NAO is the result of complex nonlinear interactions between many spatiotemporal scales (Önskog et al., 2018). The NAO can be viewed as a process with an e-folding time scale of about two weeks (Feldstein, 2000) and its life cycle may be closely linked to the anticyclonic (cyclonic) Rossby wave breaking (Franzke et al., 2004). The NAO index (NAOI) is a quantified indicator of the NAO, which is defined as the difference between normalized SLP over Iceland and Azores (Andersson, 2002).





The NAO events fall into two categories, the positive phase ($NAO^+$) and the negative phase ($NAO^-$). During the $NAO^+$, there is a strengthening of either or both the Icelandic Low and the Azores High, resulting in an increased pressure gradient, and $NAO^-$ is the opposite (Heape et al., 2013). In the past decade, the turbulence of the winter NAO has been quite extreme, and it has contributed greatly to the warm winter phenomenon throughout Europe, the cold weather in the Northwest Atlantic

(Hurrell, 1995), the dipole precipitation pattern over northwest Europe and northwest Africa (Wassenburg et al., 2016) and the surface temperature variation (Pokorná and Huth, 2015), etc. It is also found that the winter NAO is the ultimate factor affecting spring temperatures in Central Europe (Hubálek, 2016) and is closely related to the trend of warming in the northern hemisphere (Iles and Hegerl, 2017). Moreover, the multidecadal variations of the NAO can induce multidecadal variations in the Atlantic meridional overturning circulation, leading to the rapid melting of Arctic sea ice (Montreuil and Chen, 2018).

Research suggests that its seasonal variation is recognized to be mainly caused by unpredictable processes (Dunstone et al., 2016). It is, therefore, of widespread scientific research value to study the physical mechanism and enhance forecast skill for the NAO.

Although the phase and amplitude of NAO are affected by numerous factors, including sea surface temperature, anomalies in both the tropics and extratropics and stratospheric extreme events (Hansen et al., 2017), tropical atmospheric heat anomalies (Yu

and Lin, 2016) and intensity of geomagnetic activity (Bucha, 2014), the characteristics of the NAO events in the atmospherical process can be captured by the nonlinear models (Luo et al., 2007). The NAO can be regarded as a nonlinear initial value problem (Woollings et al., 2008) to explore the nature of the initial perturbation that is most likely to develop into climate events. Under some conditions, the above mentioned initial perturbation is called as the optimal precursor (OPR) (Mu et al., 2014). Such problems can be solved by conditional nonlinear optimal perturbation (CNOP), which describes the initial perturbation

that satisfies a specific constraint condition and causes the largest prediction error at the prediction time. It applies to the study of the predictability of the numerical models, simulating nonlinear motions of oceans and atmospheres (Mu et al., 2003). CNOP was originally adopted to identify the OPRs of ENSO (Duan et al., 2004), and gradually applied in research on the onset of blocking events (Mu and Jiang, 2011), Kuroshio large meander (Zhang et al., 2017b), and Indian Ocean dipole events (Mu et al., 2017b).

Very recently, Jiang *et al.* explore the optimal precursors that trigger the NAO events using CNOP, demonstrating that the amplitude induced by the self-interaction of perturbations in the onset of the $NAO^-$ is stronger than that in the onset of the $NAO^+$ (Jiang et al., 2013). On this basis, Dai *et al.* investigate the relationship between the OPR and optimally growing initial error (OGE) using CNOP (Dai et al., 2016). It is indicated that the two types of OGEs and the OPRs corresponding to the two types of NAO events have similar structures, and both of them can develop into dipole NAO anomaly patterns. These

studies provide evidence that CNOP is a useful method to investigate the onset of the NAO event. In their studies, T21L3 quasigeo-strophic global spectral model, which is a simple three-level model designed by Marshall and Molteni, is applied under ideal conditions (Marshall and Molteni, 1993). For solving CNOP, they all used spectral projected gradient 2 (SPG2) algorithm (Birgin et al., 2001), which needs the adjoint model of T21L3 to obtain the gradients of objective function for the initial condition.





The previous studies for NAO selected geopotential height as the variable to measure the events. However, as the hypothetical height in geoscience, geopotential height is often used in ideal models (Zhang et al., 2013). Besides, the traditional adjoint algorithm has its limitations in complicated operational models that do not have an adjoint available (Wang, 2010). Compared with the adjoint methods, intelligent algorithms perform better in the situation of the discontinuous objective functions (Mu
et al., 2015). In addition, the adjoint-CNOP method would fail with large initial disturbance or long prediction time due to the strong nonlinearity of the dynamical model, and local CNOPs would be produced by the adjoint-CNOP method with high probability when the objective function has multiple extreme values. In contrast, the PSO-CNOP method still achieves global CNOP and has a shorter run time in the situation of larger initial perturbations, longer prediction times, and multiple extrema values (Zheng et al., 2017). It is proved that intelligent algorithms can acquire global CNOP approximately without an adjoint
model. To enhance the performance of solving CNOP with complex numeric models, the researchers proposed intelligent algorithms based on feature extraction. The algorithms transform the problems in original input space with high dimensions into the problems in low dimension space. At present, the tentative application of intelligent algorithms based on feature extraction in solving CNOP yielded considerable achievements. The principal component analysis based genetic algorithm (PCAGA) (Zhang et al., 2017a), the Modified Artificial Bee Colony Algorithm (MABC) (Ren et al., 2016), the dynamic search
Fireworks Algorithm with linearly decreased dimension number strategy (ld-dynFWA) (Mu et al., 2017a) and PCA based Flower Pollination (PCAFP) (Yuan et al., 2016) have been successfully adopted in studying sensitive areas identification for tropical cyclone adaptive observations, El Niño-Southern Oscillation and double-gyre variation. The CNOPs obtained by these methods have similar patterns and larger fitness values in comparison to the adjoint method. It is illustrated that PCA based intelligent algorithm is appropriate for high dimensional numerical models, especially the models without the adjoint model.

The objective of this paper is to find the OPRs which produce the NAO anomaly pattern in the northern hemisphere and explore the effect of the nonlinear process. We study the case using the Community Earth System Model (CESM), which is an ocean-atmosphere coupled model without an adjoint model. In this paper, an adjoint-free parallel principal component analysis (PCA) based genetic algorithm (GA) and particle swarm optimization (PSO) hybrid algorithm (PGAPSO) is proposed to solve CNOP for NAO events. The OPRs obtained by the proposed algorithm steadily produce the SLP anomaly mode and trigger the
high NAOI. Compared against the PCA-based PSO (PPSO), the algorithm is improved to avoid falling into the local optimum and accelerates convergence. After parallelized with MPI and CUDA, the speed-up ratio of the intelligent solution system reaches $40\times$ compared with its serial version.

The structure of this paper is organized as follows: Section 2 describes the CESM, and section 3 presents the CNOP method, the PGAPSO algorithm and the parallelization technique. Experiments and results are displayed in section 4. This paper ends
with a conclusion and future work in section 5.

## 2   Community Earth System Model

The CESM (Kay et al., 2015) is a new generation of fully coupled climate models developed in 2010. It has been widely used to simulate the carbon cycle (Lehner et al., 2015), ocean currents (Large and Caron, 2015), soil moisture (Swenson and Lawrence,




2012), precipitation (Hagos et al., 2016) and other climate phenomena. As shown in Figure 1, the CESM is composed of seven geophysical model components, respectively Atmospheric (Community Atmosphere Model, CAM), Sea- ice (CICE), Land (Community Land Model, CLM), River-runoff (River Transport Model, RTM), Ocean (Parallel Ocean Program, POP), Land-ice (CISM), Ocean-wave (XWAV). The CESM also has a Coupler (CPL) that coordinates the time evolution of geophysical

models and delivers information between these components.

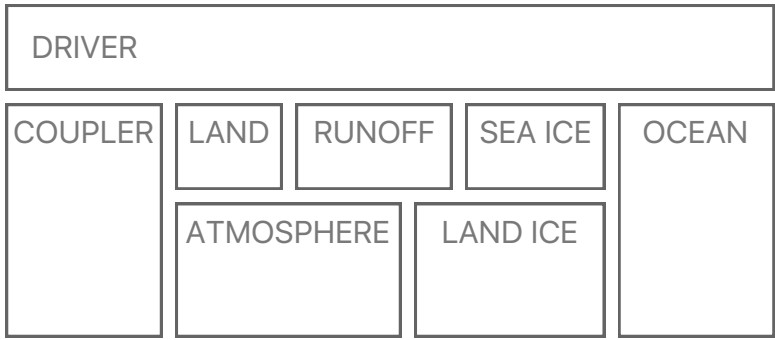

**Figure 1.** Main components of CESM.

The atmospherical component in CESM 1.2.2 is used to simulate the NAO in this work. CAM version 5.3, which is a global atmospherical general circulation model developed from the NCAR CCM3, is released as the atmosphere component of CESM 1.2. The CAM incorporates an interactive aerosol model where aerosols interact with the tropospheric chemistry. We perform the experiments on a $0.9° \times 1.25°$ horizontal grid with 26 levels in the vertical. The dataset is set to $F$ that includes CAM, CLM

and CICE(prescribed mode) activated with SST data mode. The region we focus on, which is also the NAO mainly loacated at, is a two-dimensional domain consisting of $65 \times 105$ grids with the North Atlantic area between $20°N$ and $80°N$ and between $90°W$ and $40°E$.

## 3   CNOP and PGAPSO

### 3.1   CNOP

The CNOP is a natural extension of the linear singular vector into the nonlinear regime, and is proposed to study predictability problems of weather and climate in numerical models (Mu et al., 2009). OPR is a kind of the initial perturbations that can trigger the largest uncertainty in prediction, and it can be solved by CNOP method. Specifically, the objective function achieves the maximum with the constraint condition at prediction time by superimposing OPRs on the basic state. To explore the process





of nonlinear, we choose a blocking indicator proposed by Liu (Liu, 1994) to quantify the extent of the NAO events. The NAOI is defined as the projection of the SLP field on the NAO anomaly pattern:

$$NAOI = \frac{\langle SLP_{NAO}, SLP_d \rangle}{\langle SLP_{NAO}, SLP_{NAO} \rangle} \tag{1}$$

where $SLP_d$ is obtained by subtracting the climatological mean from SLP output, and $\langle \rangle$ denotes inner product operation of

5 vectors. $SLP_{NAO}$ denotes the NAO anomaly pattern acquired by the empirical orthogonal function (EOF) analysis. The EOF is a widely used tool to decompose the spatial-temporal distribution features in geonomy (Baldwin and Dunkerton, 1999). The flow of EOF are listed as follows:

  – Process the SLP historical data into anomaly values by subtracting the mean climate state of 30-year SLP time series data, recorded as $X_{m \times n}$.

– Calculate the covariance matrix $C_{m \times m}$ via: $C_{m \times m} = \frac{1}{n} X \times X^T$.

  – Solve the eigenvalues ($\lambda_{1,...,m}$) and eigenvectors ($V_{m \times m}$) of $C_{m \times m}$ with the constraint condition: $C_{m \times m} \times V_{m \times m} = V_{m \times m} \times \Lambda_{m \times m}$.

  – The eigenvectors corresponding to $\lambda_k$ is the $k^{th}$ column of $V_{m \times m}$, that is, $EOF_k = V(:, k)$.

In general, the first mode decomposed by EOF is chosen as the NAO anomaly pattern, which is illustrated in Figure 2. The

15 NAO spatial pattern is manifested as a typical meridional dipole mode, which consists of the Iceland low pressure along with the North Atlantic subtropical high. In Figure 2, it is a positive phase of the NAO, presenting the mode with the negative anomalies in high latitude and the positive anomalies in low latitude.

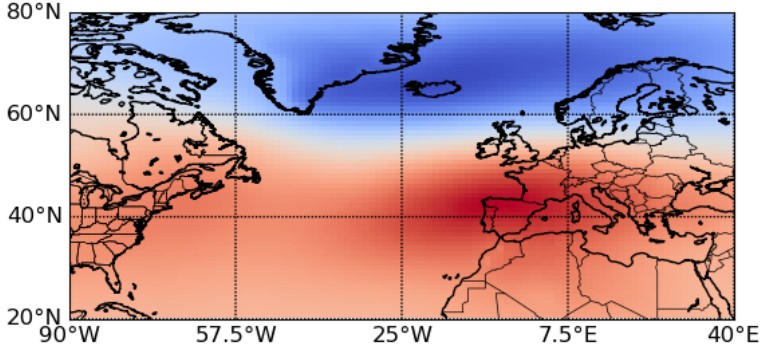

**Figure 2.** The first mode of the EOF in SLP anomaly field concentrated in the North Atlantic region between $90°W$ - $40°E$, $20°N$ - $80°N$.




The procedure for solving CNOP can be regarded as the following extrema problem:

$$J(u_0^*)_{NAO^+} = \max_{\|u_0\| \leq \sigma} J(u_0) = NAOI(NAO^+)_{CNOP} - NAOI_{refer} \tag{2}$$

$$J(u_0^*)_{NAO^-} = \min_{\|u_0\| \leq \sigma} J(u_0) = NAOI(NAO^-)_{CNOP} - NAOI_{refer} \tag{3}$$

where $u_0$ is the vector of physics variables (zonal wind, temperature, etc.). $J(u_0)$ is the objective function defined by the

difference between NAOI triggered by perturbation $u_0$ in the final state and NAOI in the reference state. According to the formula (2) and (3), the perturbation $u_0^*(NAO^+)$ makes $J(u_0)$ achieve the maximum, whereas $u_0^*(NAO^-)$ makes $J(u_0)$ reach the minimum. They are the initial perturbations defined as OPRs. $\sigma$ denotes the constraint condition of the OPRs. We reference the constraint condition in a similar study in the field of the atmosphere. In the study of the identification of the sensitive areas for tropical cyclone using CNOP, the objective function is chosen as the summation of kinetic energy, available

relative potential and surface potential energy in the verification areas D (Zhang et al., 2017a):

$$J(u_0)_{Trop} = \frac{1}{D} \int_D \int_0^1 [u'^2 + v'^2 + \frac{C_p}{T_r}t'^2 + R_a T_r (\frac{\pi'}{\pi_r})^2] d\sigma dD \tag{4}$$

where $u'$, $v'$, $t'$ and $\pi'$ are initial perturbations. $C_p$ is the specific heat at the constant pressure which is set to 1005.7 $J \cdot kg^{-1}K^{-1}$ and $T_r$ is the reference temperature with a value of $270K$. $R_a$ denotes the ideal gas constant, and its value is set to 287.05 $J \cdot kg^{-1}K^{-1}$. $\pi_r$ is the reference static pressure with a value of 1000 $hPa$. In order to ensure the perturbations within

a reasonable range, the constraint is set to 10% of the dry energy norm in the basic state, that is:

$$\sigma = 10\% * \frac{1}{D} \int_D \int_0^1 [U_0^2 + V_0^2 + \frac{C_p}{T_r}T_0^2 + R_a T_r (\frac{\Pi}{\pi_r})^2] d\sigma dD \tag{5}$$

We adopt the above constraint $\sigma$ since the physical variables are the same with the variables in research on the tropical cyclone. Combining the formula (1), (2) and (3), the objective function is described as follows:

$$J(u_0) = \Delta NAOI$$
$$= \frac{\langle M_{t_0 \to T}(U_0 + u_0) - M_{t_0 \to T}(U_0), SLP_{NAO} \rangle}{\langle SLP_{NAO}, SLP_{NAO} \rangle} \tag{6}$$

where $M_{t_0 \to T}$ represents the nonlinear model propagator from initial time $t_0$ to the prediction time T, and $U_0$ denotes the initial basic state. Therefore, $M_{t_0 \to T}(U_0)$ denotes the reference state at prediction time T. The objective function is the projection of SLP field difference between the final state and the basic state on the NAO anomaly pattern.



## 3.2 PGAPSO

Under the resolution of f09_g16 with an approximate grid spacing of $0.9° \times 1.25°$, the total dimensions of variables involved in the objective function are 5861376. It is difficult for the algorithm to solve the optimization problem in such high dimensions. Thus, we need to extract the feature of samples to reduce the data scale.

PCA is a traditional method for feature extraction and has been widely used in signal separation (Kasban et al., 2016), environment forecasting (ULSAUFIE et al., 2013) and pattern classification (Li et al., 2017), etc. In this paper, we adopt PCA to implement dimension reduction for sample data. The original winter sample is generated by running 10-year integration using CESM. Then subtract the climatological mean of the above integration from the original data, and the obtained sample is weighted according to the area of the grid:

$$S_i = (S_i - \frac{1}{n}\sum S_i) * \cos(lat_i) \, (i = 1, 2, \dots, n) \tag{7}$$

where $lat_i$ is the latitude of the $i^{th}$ row in the grid, and the weight is calculated approximately via the cosine value of the grid's latitude. Then the eigenvalues $(\lambda_1, \dots, \lambda_n)$ and eigenvectors of the covariance matrix $SS^T$ are calculated to obtain principal components:

$$SS^T L = L\Sigma \tag{8}$$

The top $m$ columns of the eigenvectors $L$ sorting by their eigenvalues are selected as the principal components. The value of $m$ is determined by the contribution rate, which is defined as:

$$r = \frac{\sum_{i=1}^{m} \lambda_i}{\sum_{i=1}^{n} \lambda_i} \tag{9}$$

In this work, $m$ is set to the minimum number of columns that meet the contribution rate of 95%. The reduced space with $m$ dimensions is far smaller than the original one.

To obtain the extremum of the objective function, we adopt a hybrid algorithm improved from two classical algorithms, PSO and GA. The PSO is a type of intelligent heuristic algorithm to solve the problem with NP property (Kennedy, 2011). The position with the best fitness value is searched by tracing individual optimal positions and the optimal global position in the meantime. The flow of the algorithm is described in brief: (1) Initialize the speed ($V$) and position ($X$) of particle swarm with random values. (2) For each particle $i$, the position vectors in reduced space need to be restored into original space via

$X_i' = X_i \cdot L_{1,\dots,m}$, and superpose the perturbation $X_i'$ on the basic state. When the model integration is finished, calculate





the fitness value of each particle through the formula (6) and record its optimal position ($X_{pb}$) along with the global optimal position ($X_{gb}$). (3) Update position and speed of each particle. The updating formula is as follows:

$$\begin{cases} V_i^{k+1} = \omega_k V_i^k + c_1 r_1 (X_{pb}^k - X_i^k) + c_2 r_2 (X_{gb}^k - X_i^k) \\ X_i^{k+1} = X_i^k + V_i^{k+1} \end{cases} \tag{10}$$

where $V_i^k$ is the speed of particle $i$ for step $k$ and $V_i^{k+1}$ is for step $k+1$. $c_1$ is the self-awareness coefficient for the historical

self-optimal position and $c_2$ is the social-awareness coefficient for global optimal position of all particles. The empirical value of $c_1$ and $c_2$ are both set to 2. $r_1$ and $r_2$ are random float numbers with uniform distribution in [0, 1]. $X_{pb}^k$ refers to the position of particle $i$ where objective function acquires the maximum(minimum) in $k$ steps, and $X_{gb}^k$ represents the position where objective function achieves global extrema in $k$ turns. Both position vectors and speed vectors are in reduced space with $m$ dimensions. $\omega_k$ is the weight parameter and calculated by:

$$\omega_k = \omega_{max} - \frac{\omega_{max} - \omega_{min}}{iter_{max}} * iter \tag{11}$$

where $iter$ is the current number of step, and $iter_{max}$ is set to 100.

In PGAPSO, PSO is viewed as the main body of the search process, and the GA further optimizes the position. As a meta-heuristic algorithm, the GA derives from natural selection (Goldberg and Holland, 1988). When the fitness value is obtained in step (2) of PSO, the particles are fed into GA for further search. A portion of particles in the existing population are selected

according to their fitness value to breed a new generation. The selection operation is performed on roulette strategy, that is to say, the probability that each individual is selected is equal to the ratio of its fitness value to the total fitness value of the entire population:

$$p_s = \frac{J(u_{X_i'})}{\Sigma J(u_{X'})} \tag{12}$$

After that, the selected parents generate new individuals with crossover:

$$\begin{aligned} X_a'\{x_s,\dots,x_e\} = X_b\{x_s,\dots,x_e\} \\ X_b'\{x_s,\dots,x_e\} = X_a\{x_s,\dots,x_e\} \end{aligned} \tag{13}$$

Then the new generation mutates with probability $p_m$ in a single position to avoid genetic drift. The fitness value of each new generation is compared against its parents, and the best position is recorded. With the optimal local position and global optimal position, the speed and position of particles are updated using formula (10). The final global fitness value is obtained until the $iter$ reaches $iter_max$ or the norm of particles' speed approaches zero.





### 3.3 Parallelization

The computation of CNOP in CESM is quite time-consuming. With 48 CPU cores, 30 particles and 100 iterations, it takes about 13.75 days to obtain the OPRs in the serial program. For PGAPSO to operate more effectively, multiple parallel techniques and frameworks are adopted in this work.

### 3.3.1 CESM Parallelization

The role of the CAM component in CESM is to simulate the variation of atmosphere and ocean, and the largest variation can be discovered by objective function using PGAPSO. With high resolution, the input data handled for integration in nonlinear processes of CAM possess the features as massive variables, high dimensions and complexity, which makes the invocation for CAM become the primary time-consuming task in the whole program. Although CESM has already been parallelized using Message Passing Interface (MPI) and Open Multi-Processing (OpenMP), it is still time-consuming.

Recently, the Graphics Processing Unit (GPU) has been widely used in accelerating numerical models. Since GPU is suitable for parallel computing on a large scale, it can significantly improve the execution performance of climate models. A parallel scheme for Community Climate System Model (CCSM) has been proposed to shorten the runtime of climate prediction by porting the radiation module onto GPUs (Coleman and Feldman, 2013). The module was parallelized using the inline method and communicated with MPI routines. A cloud analysis scheme called Goddard Cumulus Ensemble (GCE) in Weather Research and Forecasting (WRF) was highly expedited using NVIDIA Tesla K40 with 2880 cores (Huang et al., 2015). Compared to CPU-based parallel version running on 4 nodes, the GPU-based scheme performed faster. As for CESM, the novel asynchronous execution strategy has provided significant performance benefits (Korwar et al., 2013). The most time-consuming routines have been accelerated via OpenACC directives and achieved a speedup of $1.19\times$-$1.53\times$ for the entire model. Another attempt for accelerating CESM was to port CESM along with a rewritten vertical remapping scheme onto GPUs (Carpenter et al., 2013). The results indicated that the performance of the optimized subroutine was improved substantially. Related works show that GPU is an alternative approach to enhance the performance of the climate model.

In this work, we port several time-consuming subroutines in CAM onto GPUs through PGI CUDA Fortran interface. After analysis run time using *pref*, shown in Figure 3, subroutine *radclwmx* and *radabs* both consume longer runtime compared with other subroutines. These two subroutines are both optimized with CUDA platform. Simultaneously, kernel directives and OpenACC directives are used to implement simplification of specific operations on the device. To reduce transfer latency between host and device, asynchronous streams are overlapped calculation with data transmission. Moreover, the loops in subroutines are merged and reconstructed to minimized I/O transfer times. During the compilation phase, the compiler option $-O4$ is selected to perform the optimization of the highest level. The command *-fast* and *-fastsse* are also utilized to launch the 64-bit Single Instruction Multiple Data (SIMD) instruction and implement cache alignment and flush.



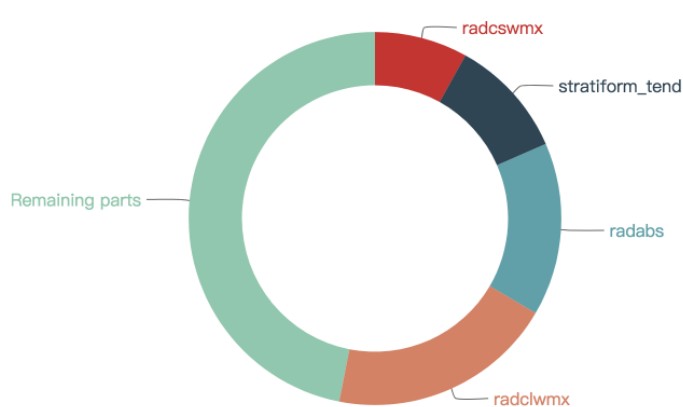

**Figure 3.** Execution time division of CAM routines.

### 3.3.2 PGAPSO Parallelization

In the process of solving CNOP using PGAPSO, the calculation of fitness value for each particle in each iteration is relatively independent. Thus it is suitable for multi-process techniques to execute these tasks concurrently. Here we adopt MPI as the parallel framework to accelerate the algorithm. MPI enables parallelization of the program via launching multiple processes with supporting communication and broadcasting between nodes. Assume that $n$ particles are initialized, then we assign one process for each particle so that the objective function along with the climate model can be called in parallel. Then the master process compares the current particles according to fitness value, and the position and speed of particles are updated at the end of each iteration. With the help of MPI, the performance of PGAPSO can be significantly enhanced.

The flow of the parallel PGAPSO is described by the pseudocode:

Figure 4 demonstrates the parallel architecture of PGAPSO for solving CNOP. The processes are divided into two groups: the master process and slave processes. At each iteration, the master process allocates calculation tasks to slave processes. For each process, perturbations under constraint condition are superimposed on CESM. Then CESM, which is paralleled with MPI, OpenMP and CUDA, is called to perform the integration. The fitness values of each process are calculated by projecting SLP output on NAO anomaly pattern. When all the fitness values are acquired, the master process gathers the fitness values from slave processes and broadcasts the optimal global value to slave processes via MPI. Then the crossover and mutation operations are performed if the norm of particles' speed is less than the threshold value.





---

**Algorithm 1** Pseudo code for PGAPSO

1.5

1: transform training data through PCA to obtain principal components $L_d$ with $m$ dimensions

2: initialize population

3: **for** $iter = 1$ to $iter_{max}$ **do**

4:     **for** each process **do**

5:         restore solution matrix $X_i$ into original space via $X_i * L_d$

6:         calculate fitness value $J(u_0) = F(\|M_{t_0 \to T}(U_0 + P * L_d)\|)$ under the constraint $\sigma$

7:     **end for**

8:     gather results from each process

9:     **if** norm of particle speed $\leq \xi$ **then**

10:         select individuals according to $\frac{J(u_{x_i'})}{\Sigma J(u_{x_i'})}$

11:         crossover and mutate with probability $p_m$

12:         compare fitness values between new generation and parent individuals

13:     **end if**

14:     update particle speed via $V_i^{k+1} = \omega_{i0} V_i^k + c_1 r_1 (X_{pb}^k - X_i^k) + c_2 r_2 (X_{gb}^k - X_i^k)$

15:     update particle position via $X_i^{k+1} = X_i^k + V_i^{k+1}$

16: **end for**

---

## 4 Experiments and Results

### 4.1 Experimental Environment

We conduct experiments on Tianhe-2 supercomputer, which is located in the National Supercomputer Center in Guangzhou, China. Each node consists of 2 Intel Ivy Bridge Xeon processors connected by Intel QuickPath Interconnect. NVIDIA Tesla
5  K80 GPUs on Tianhe-2 are used in our GPU-based scheme for CESM acceleration. Each Tesla K80 GPU has 4992 CUDA cores, and its double-precision performance is up to 2.91TFLOPS. Data transmission between CPUs and GPUs depends on PCI-e 3.0 bus with 40 lanes.

### 4.2 Experimental Procedures and Results

The first step is to decompose the principal component from the original sample. We run a 10-year integration (only in winter)
10  using CESM, and the samples are obtained by subtracting the winter climatological mean to eliminate the linear correlation. The dimension of the principal component is determined by the cumulative variance proportion. Table 1 reports the cumulative variance proportion at a different number of eigenvalue. The sum of variance proportion increases as the dimension of principal components increases. To balance the computation result and performance, we select the top 50 eigenvectors as the principal components corresponding to the cumulative explained variance ratio of 95%.




**Figure 4.** The parallel architecture of PGAPSO for solving CNOP.

**Table 1.** The variance proportion at different number of eigenvalues.

| Number of eigenvalue | 10 | 20 | 30 | 40 | 50 | 60 |
|---|---|---|---|---|---|---|
| Variance ratio | 82.05% | 89.70% | 92.66% | 94.34% | 95.45% | 96.21% |





The purpose of this paper is to explore the mechanism of the nonlinear system and to find which type of perturbation can trigger NAO events. In other words, we aim to find out the OPR for NAO events. According to the definition of OPR, OPR is the state most likely to lead to the pattern with the highest NAOI. The incremental value of the NAOI, which is defined by $NAOI_{pert} - NAOI_{refer}$, is the way to measure the extent of the NAO events. Thereinto, $NAOI_{refer}$ stands for the final

NAOI acquired without perturbations at prediction time. Since the time scale of NAO events is about two weeks, we select 5 days, 7 days and 15 days as the simulation time to observe the variation of index amplitude. The number of particles is set to 30, and the times of iteration is set to 100. The perturbations are superimposing on the Arctic region ($60°N$ - $90°N$) consist of six variables which are listed in Table 2. Here we adopt $CNOP_{PO}$ and $CNOP_{NE}$ to express the OPRs corresponding to the positive-phase NAO and the negative-phase NAO respectively. Figure 5 displays the trends of the NAOI amplitude

for $CNOP_{PO}$ (red line), $CNOP_{NE}$ (blue line) and reference flow (black dashed line). Figure 5 portrays the change of the index for the reference state and perturbation state. As can be seen from the diagram, the reference state flow fluctuates on a small scale and sustains positive value. The $CNOP_{PO}$ and the $CNOP_{NE}$ both contribute to a high anomaly index state, and $|\Delta NAOI(-)|$ is significantly greater than $|\Delta NAOI(+)|$. In the final days of the simulation time, rapid variation occurs, and the increment value reaches greater than 1 or less than -1. It is illustrated that the nonlinear process plays a role mainly on the

last stage of the evolutionary period.

**Table 2.** The related variables included in the perturbations.

| Variable name | Description | Units |
|---|---|---|
| U | Zonal wind | $m/s$ |
| V | Meridional wind | $m/s$ |
| T | Temperature | $K$ |
| Q | Specific humidity | $kg/kg$ |
| PS | Surface pressure | $Pa$ |
| PHIS | Surface geopotential | $m^2/s^2$ |

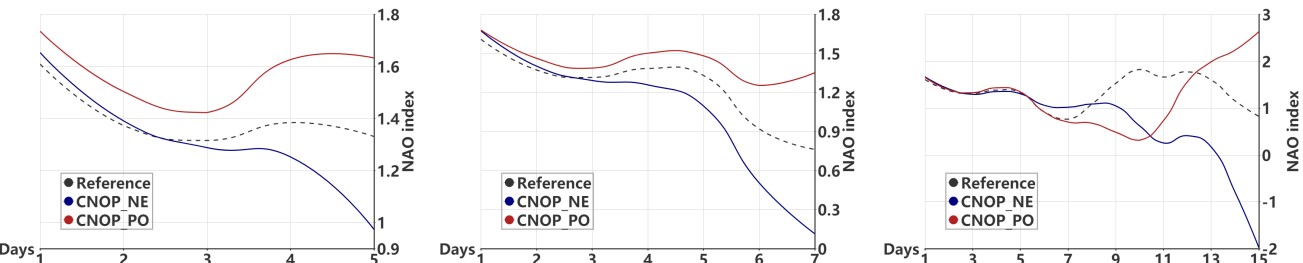

**Figure 5.** The trends of the index amplitude for $CNOP_{PO}$, $CNOP_{NE}$ and the reference flow.





To check the effectiveness of CNOP method, random perturbations are generated and superposed on the basic state for comparison. With the same start date and iteration number, the NAOI evolution of random perturbations and the NAOI evolution of CNOPs are displayed in Figure 6. 15 random particles are chosen for illustration in this figure. It is demonstrated that the NAOI of perturbation state for CNOPs (red line for $CNOP_{PO}$ flow and blue line for $CNOP_{NE}$ flow) have the largest growth compared against random perturbations (grey lines). The evolution tendencies of these random perturbations are similar to the reference flow in Figure 5, and are not well-distribution. In addition, the results of random perturbations are located around the reference state with a large probability. Thus, it is important to apply intelligent algorithms to find the search direction. By optimizing with PGAPSO and CNOP method, the final state of $CNOP_{PO}$ achieves the maximum value of the NAOI, and the $CNOP_{NE}$ achieves the minimum, shown in each subgraph of Figure 6.

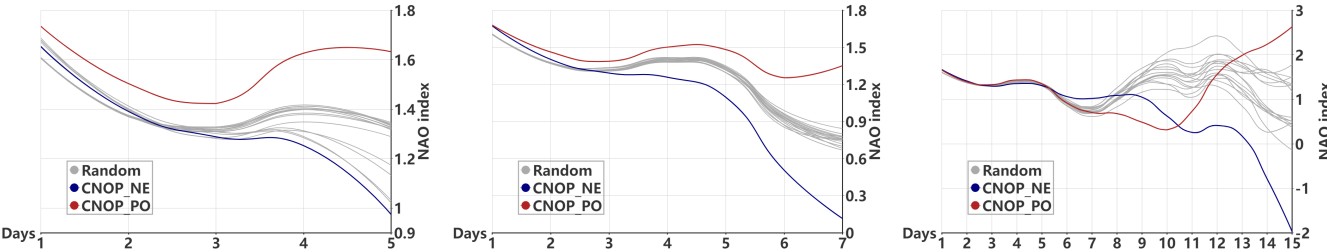

**Figure 6.** The evolution of NAOI with the simulation time of 5 days, 7 days and 15 days.

To evaluate the NAOI of CNOPs more visually, Table 3 reports the incremental values of the NAOI with different simulation time. From Table 3, the difference between the NAOI in the final state and the NAOI in reference state increases when the integration time becomes longer. With simulation time of 15 days, $|\Delta NAOI|$ is far greater than 1.

**Table 3.** The increment value of NAOI with different simulation time.

| Simulation time | $\Delta NAOI^+$ | $\Delta NAOI^-$ |
|---|---|---|
| 5 days | 0.3 | -0.36 |
| 7 days | 0.59 | -0.65 |
| 15 days | 1.81 | -2.80 |

Figure 7 shows the two types of SLP patterns triggered by OPRs for 5 days, 7 days and 15 days. The left column displays the positive phase, and the right column displays the negative phase. The SLP field is obtained by $SLP_{pert} - SLP_{refer}$, and the $SLP_{refer}$ denotes the final SLP field without superimposing the perturbations. The typical pattern of NAO is the dipole mode located near Iceland and Azores. For 5-day optimization, several positive centers and negative centers are concomitant and overlapping in the region, with the negative (positive) core arising at the north of $60°N$. In the evolution of 7 days, positive (negative) cores move to the position around the Davis Strait and western Europe. Both of the 5-day final state and 7-day final state haven't developed into NAO events. The dipole mode, which has a strong negative (positive) pressure center situates on


Iceland with positive (negative) SLP field over the middle latitudes of the North Atlantic Ocean, forms in a 15-day integration. Under the influence of the nonlinear process, the dipole centers migrate across the Atlantic Ocean.

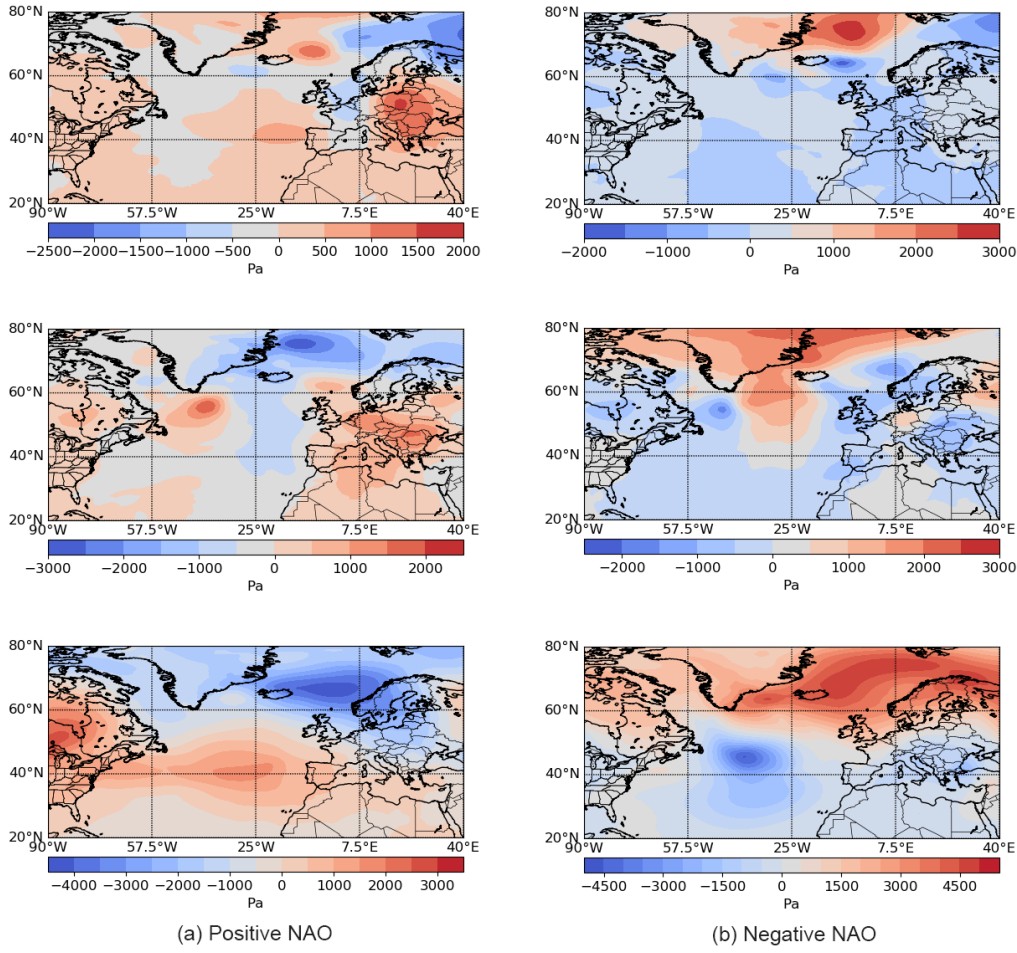

(a) Positive NAO    (b) Negative NAO

**Figure 7.** The $NAO^+$ (a) ($NAO^-$ (b)) mode at SLP field (Pa) triggered by $CNOP_{PO}$ ($CNOP_{NE}$) with the simulation time of 5 days, 7 days and 15 days.

In summary, the difference between $NAOI_{CNOP_{PO}}$ and $NAOI_{CNOP_{NE}}$ increases when simulation time growth within 15 days, and the increment value of NAOI in 15-day optimization reaches the maximum. In Figure 7, the 15-day integration forms the typical NAO pattern. These above diagrams demonstrate that the 15-day optimization can evidently trigger the NAO events. Besides, under the action of CNOPs, the basic state can evolve into both positive and negative phase of the NAO events. Therefore, we choose 15 days to perform model integration in this case.



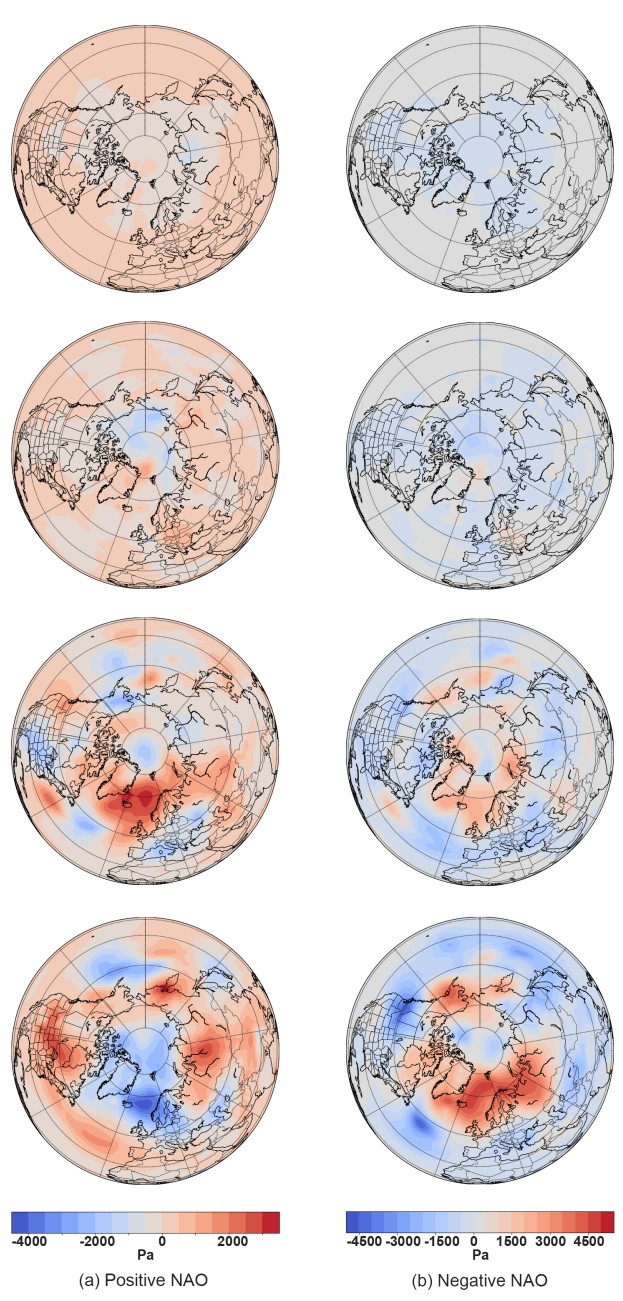

(a) Positive NAO          (b) Negative NAO

**Figure 8.** The nonlinear evolutions on day 1, day 5, day 10 and day 15 at SLP field (Pa) with an simulation time of 15 days.




To observe the evolution process of NAO events in 15 days, we plot the SLP field on day 1, day 5, day 10 and day 15. For $NAO^-$ in the right column, the negative-pressure difference increases through the whole process, with positive-pressure difference occur and strengthen during day 10-15. The basic dipole structure forms on day 10, and gradually develop into NAO anomaly event. As for $NAO^+$, with a complicated process, a strong positive center locates on the Norwegian Sea on day 10,

compensating the enhanced southward oceanic heat transport. On day 15, the sense of the gyre will change sign to become negative on the region we focused, and develop into an anomalous pattern. It is consistent with the right subgraph of Figure 5, showing that the NAOI of $CNOP_{NE}$ is under a sustained downward trend. For $CNOP_{PO}$, the NAOI sinks to the lowest point on day 10 then rises quickly in the final 5 days.

The above evolutions are triggered by superimposing perturbations on the Arctic region with multiple variables. We find out

that the NAO events can also be triggered by a single variable, like temperature. Following the above procedure, the temperature perturbations are limited under a constrained condition of $T^{'2} \leq 100$ and superimposing on the $25^{th}$ level of atmosphere (near surface) in the same region. By using PGAPSO, the NAOI converge to optimized values. The perturbations are illustrated in Figure 9. As seen in Figure 9, the phase of $CNOP_{PO}$ and $CNOP_{NE}$ has an almost opposite structure in the North Atlantic sector. There exist an obvious pressure difference between Greenland and Iceland, with several centers in the mid-to-high

latitudes and small cores around the Arctic region. Besides, the positive anomaly in eastern Europe is also conducive to the formation of the dipole. It matches up with the hypotheses that atmospherical temperature gradients will result in the anomalous poleward atmospherical heat transport and an increased probability of the NAO occupying its high index state.

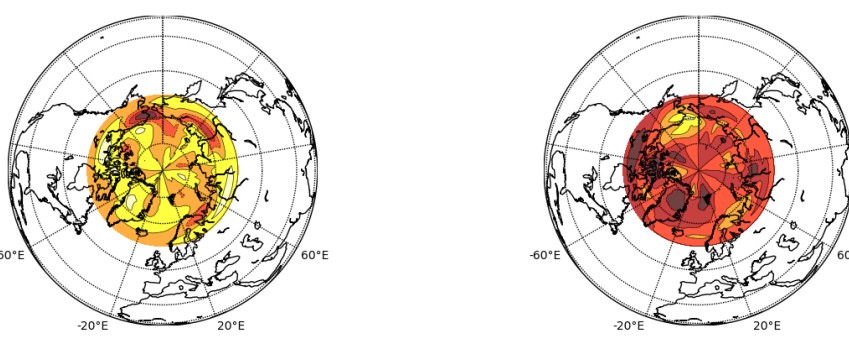

**Figure 9.** The temperature perturbations ($^{\circ}$C) superimposing on the Arctic region which trigger the NAO events

### 4.3 Performance Analysis

In order to demonstrate the performance improvement of parallel PGAPSO adopted in this paper, Figure 4 compares the

runtime of parallel PGAPSO and serial PGAPSO for one iteration. The runtime of CESM is the performance bottleneck of the algorithm, which can be broken by running in parallel. Our parallel scheme using MPI implements the simultaneous execution of multiple particles to solve the problem. From Figure 4, we can see that when the number of CPU cores is more than 840,





it will take longer to run the serial algorithm. Since CESM has been paralleled with MPI and OpenMP, when the number of CPU cores increases to the critical point, the time of communication between nodes makes the increase of the CESM runtime. The speedup ratio of parallel PGAPSO compared with serial PGAPSO is displayed in Table 4. The speedup ratio increases with the rise of the CPU cores' number. With assigning CPU cores to multiple tasks, the execution time of parallel PGAPSO

continues to decline, while the serial PGAPSO takes longer owing to communication. With 1080 CPU cores, PGAPSO based on the parallel scheme achieves a speedup of $40\times$ compared to its serial version.

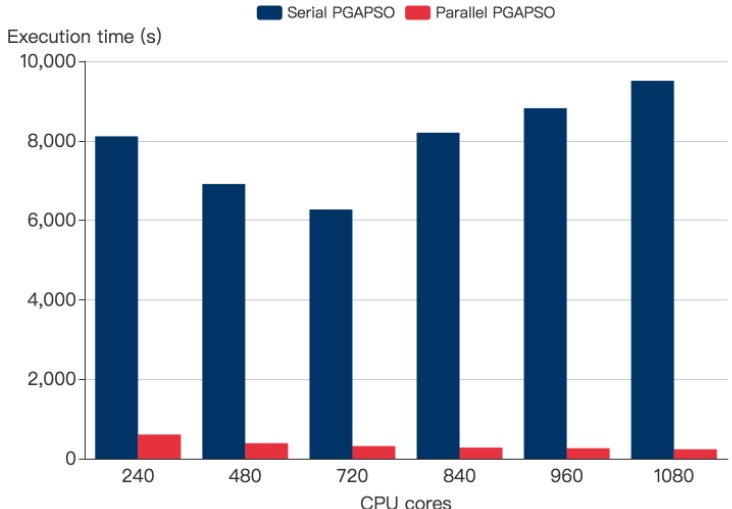

**Figure 10.** The execution time of serial PGAPSO and parallel PGAPSO with different number of CPU cores.

**Table 4.** The speedup of parallel PGAPSO compared with serial PGAPSO.

| Number of CPU cores | 240 | 480 | 720 | 840 | 960 | 1080 |
|---|---|---|---|---|---|---|
| Speedup ratio | $13.3\times$ | $17.6\times$ | $19.5\times$ | $29.3\times$ | $33.4\times$ | $40.0\times$ |

The parallel PGAPSO with GPU technique has also compared against the parallel PPSO and the PGAPSO, which are parallelized without accelerating CESM. Figure 11 shows how long each version runs in one iteration. From Figure 11, along with the increase of CPU cores, all parallel methods have a trend of decrease in time consumption. When the CPU cores are

10 increased to 1080, the runtime of these three methods for one step is 335.64s, 251.88s and 238.17s respectively. PGAPSO takes slightly longer to execute compared with PPSO since offspring particles are generated and calculate the objective value with probabilities. In these procedures, integration of CESM is the most time-consuming part, which becomes a source of the performance bottleneck. The accelerators in GPUs enhance the performance of the nested loop in the atmosphere component. Although only two subroutines are accelerated in this work, PGAPSO obtains considerable speedup effects combined with

15 GPUs. It is also demonstrated the potential capacity of GPUs in accelerating numerical models.



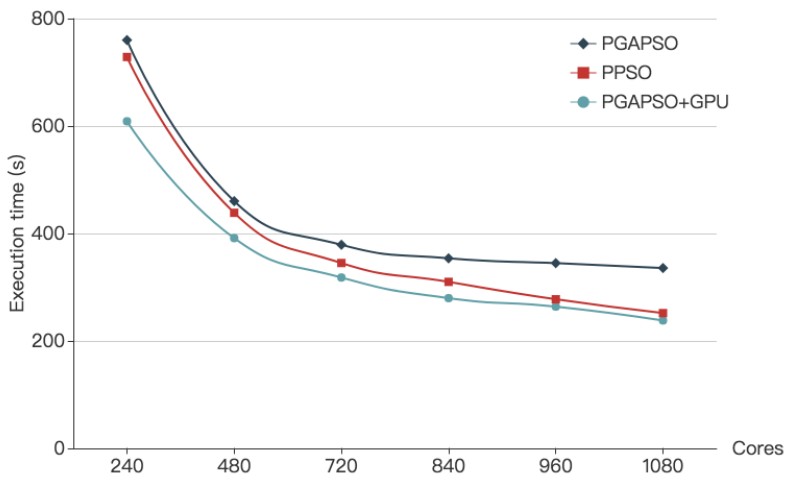

**Figure 11.** Time consumption of PGAPSO, PPSO and PGAPSO(GPU) with the growth of CPU cores.

Meanwhile, the convergence and optimal values of PGAPSO are also compared with PPSO, shown in Figure 12. The speed norm is a representation of position offset between the current particle and optimal particle. When the speed norm approaches zero, all particles move to an adjacent site which the algorithm converges to. The speed norm in PGAPSO falls rapidly around $10^{th}$ step, and converges in about step 30. The absolute value of increment NAOI for PGAPSO, which is

expressed as $|\delta NAOI|_{PGAPSO}$, is greater than PPSO. The advantages of the hybrid algorithm show in two perspectives: the crossover operation of GA has the relative larger probabilities of generating the generation with higher fitness value since the particle parents are selected according to the proportion of fitness value. Besides, the mutation operation increases the randomness of the current particle to avoid plunging into local optima. Thus, PGAPSO improves the convergence rate and solution quality compared with PPSO.

Moreover, the standard deviation is used to measure the stability of the PGAPSO, and is shown in Figure 13. By testing the PGAPSO in 10 times, the standard deviation of 5-day optimization and 7-day optimization are both less than 0.05. Owing to the more considerable fitness value, the 15-day result is relatively more significant. Overall, the PGAPSO is reliable.

## 5   Conclusions

To improve the predictability of the NAO, we adopt a CNOP-based approach for the exploration of the NAO's optimal precur-

sors. Since the CESM does not have corresponding adjoint models, we cannot solve CNOP through ADJ-based method in the works of predecessors, such as SQP and SPG2. In this paper, we propose a parallel PCA-based hybrid algorithm which coupled PSO and GA (PGAPSO) to solve CNOP in CESM. As an adjoint-free method, PGAPSO effectively solves the problem in exploring initial perturbations that cause the NAO events. In the process of iteration, the CESM is regarded as a black box




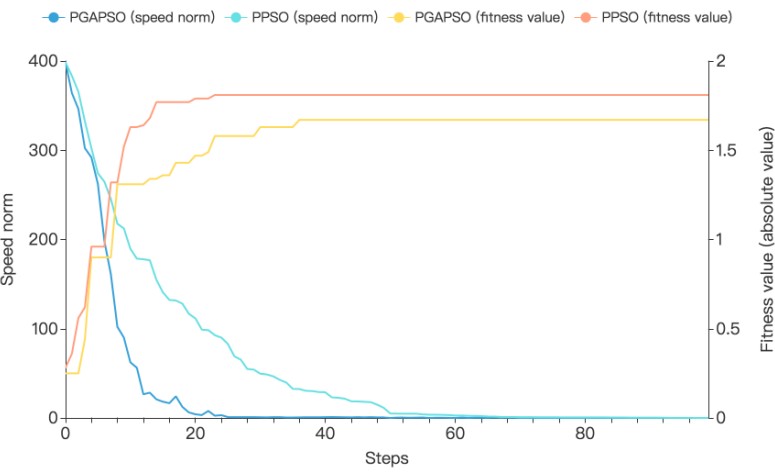

**Figure 12.** The speed norm of particles and global optimal fitness value of PGAPSO and PPSO in 100 steps.

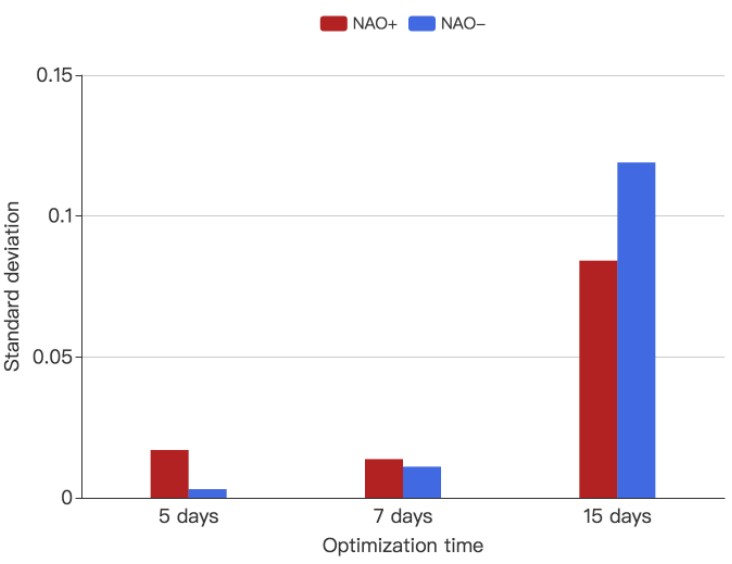

**Figure 13.** The standard deviation of NAOI obtained by PGAPSO with simulation time of 5 days, 7 days and 15days.





program. It is convenient to transplant the solver framework to other numerical models. Moreover, the parallelization mainly consists of two parts: parallelization of the algorithm with MPI and acceleration of CESM using CUDA. It observably enhances the performance of its sequential version and achieves a speed-up of $40.0\times$. Our future work is to apply the PGAPSO algorithm to study of other climatological phenomena with CNOP method. We will also apply our approach to models which have high

5   dimensions and have no corresponding adjoint model.

*Author contributions.* Jing Li wrote the original manuscript and designed the algorithm and parallel scheme; Xiaodan Luo and Jing Li configured the simulation environment and performed the experiments; Bin Mu and Shijin Yuan supported the project, reviewed and edited the manuscript. We also thank the help of Dai as an expert in the area of atmosphere.

*Acknowledgements.* This work was supported by National Natural Science Foundation of China with grant number [41405097]. The calcu-

10   lation of this work was performed on Tianhe-2. Thanks for the support of National Supercomputer Center in Guangzhou (NSCC-GZ).





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
