# Peer review of "A Parallel Hybrid Intelligence Algorithm for Solving Conditional Nonlinear Optimal Perturbation to Identify Optimal Precursors of North Atlantic Oscillation"

_Nonlinear Processes in Geophysics, 2019_

## Referee Comment (RC1) · Anonymous Referee #1 · 24 Jun 2019

The paper describes a refined algorithm for identifying the optimal precursors (OPR:s) of the North Atlantic Oscillation (NAO).

The focus of the article is somewhat unclear, focusing both on the speedup of the computations and on the meteorological results obtained. There is a clear introduction in which the methods and problems investigated are put into the proper context with plenty of relevant citations, but I would like the authors to put also the results obtained into context, in particular by comparing them to the results of Jiang et al. (2013) and Dai et al. (2016). I would also like the author to highlight which of the algorithms that

are their original contribution.

I would also like the authors to include a section where it is explained what the usefulness of the OPR:s are. Are these states stable in the sense that an anomaly pattern that is close to the OPR (in some sense) is more likely to cause an extreme NAO event in 15 days than other patterns, so that the OPR:s can be used to make predictions about the future NAO state. The OPR:s are found by maximizing the error at the terminal date, but is the nonlinear process reversible in the sense that the OPR:s give us information about the future development of the NAO. A statistical investigation of this connection would be an interesting expansion of the article.

The equations in the article are not very well written and a number of variables and notations are not defined in the text. Some examples of these flaws are given below: Page 5, line 9: What is m and n? Page 6, equations (10) and (11). The choice of notation is not very good and it is unclear how it realtes to the notation $u\_0^{*}(NAO^+)$ used on line 6 of the same page. Page 6, equation(4): What is "Trop" and what are u', v', t' and pi' perturbations of? Page 6, line 20: What is meant by "nonlinear model propagator"? Page 7, equation (7): What is $S\_i$ here, what set is the sum taken over, what is n here? Use lat(i) instead of lat_i. If lat_i is the latitude of the ith row then, i should only take 65 different values. Page 7, equation (8): What is L and Sigma? Page 7, line 24: How are the ranodm values of V and X chosen? Page 7, line 25: Is $L\_{1,...,m}$ an m times m diagnoal matrix with the first m eigenvalues?

Minor comments: Page 4, line 18: What is meant by "To explore the process of nonlinear,..."? Page 18, lines 11-12: What is meant by "...are generated and calculate the objective value with probabilities".
* * *

---

## Editor Comment (EC1) · Balasubramanya Nadiga (Editor) · 25 Jul 2019

A heuristic optimization procedure is used to identify finite amplitude initial (day 0) perturbations that maximize an NAO related index a few days later (5, 7, and 15). A few comments/suggestions are offered below.

A cursory google search shows that the heuristic optimization procedure that is being proposed here has been previously proposed. Please clarify, discuss and cite related literature.

If individuals in PSO are being mutated/crossed-over in the GA phase of the hybrid algorithm, what is the logic in retaining the original updating rules (Eq. 10) for PSO that depend on the individual's best position?

Please state clearly what is being done for the multiple variables that are being used in the optimization procedure with respect to PCA; are multivariate PCAs being used? if not likely different variables will require different numbers of components to reach the 95% captured variance. How are the different variables weighted?

It would seem that the step corresponding to "compare parents and offspring" requires computing fitness values which in turn requires invoking CESM

Make notation consistent in figure. $||V\_x||\_2 <= \backslash xi$

Is the reference flow the same in the three panels of Fig. 5? Seems so. If so, mention it explicitly. How was the initial reference state chosen. Where multiple initial reference states considered. Do the results presented depend on the initial reference state?

Since the anomalies are increasing with simulation time for the simulation times considered, yet longer simulation times should be considered to better characterize this relationship.

Please specify how the amplitudes of the random perturbations were chosen in the context of Fig. 6

How do the serial OPRs compare with the parallel OPRs?

What is the dependence of the OPRs on initial random seeding? This characterization seems important to establish significance of the paper's findings given the heuristic nature of the algorithm.

Practitioners/interested readers may find the inclusion of a brief discussion of CNOP OPRs in the context of more commonly used initial perturbations such as singular vectors and bred vectors useful.

---

## Referee Comment (RC2) · Anonymous Referee #2 · 12 Aug 2019

In this study, authors propose a hybrid algorithm to identify atmospheric perturbation that could result in an extreme NAO event in the foreseeable future. The technique is demonstrated using CESM simulation. The performance of the algorithm as well as of the CESM simulation is enhanced with the help of MPI and CUDA tools. The authors' original contribution, however, is not clear. The language used in the text is not up to the standard required for publication. Overall, the study merits publication and further scrutiny by broader audience after following revisions.

page 3, line 1-2 : 'However, as the hypothetical height in geoscience, geopotential

height is often used in ideal models'. Not true! It is used many complex models as well as observational analysis. It is not clear to me how is this relevant to current discussion.

page 3, line 1-25 : Please clearly state your original contribution and differentiate it from previous works.

page 3-4, section 2 : Please add simulation details and describe the data in terms of variables used, their frequency etc. It would be better to explain these rather than describing CESM and its components. It is not clear to me how was the atmospheric model forced at the surface.

page 5, line 1: Does NAOI stands for NAO index?

page 5, line 10: Please state what m and n are?

page 6, line 8-18 : Please provide more discussion on why is it alright to use the same constraint (equation 5) as the one used for identifying sensitive areas for tropical cyclones.

page 7, line 7-8 : It is not very clear how was the original winter sample generated. Also on page 11, line 9.

page 8, line 25 : correct 'iter_max'

page 9, line 6 : CAM component doesn't simulate or prognose ocean variations

page 9, line 27 : 'asynchronous streams are overlapped calculation with data transmission'. It doesn't look like a correct usage.

page 17, line 19-22 : incorrect figure reference; Figure 4 instead of Figure 10

page 18, line 1 : parallelized is more commonly used in this context.

page 18 line 2 : '...the time of communication between nodes makes the increase of the CESM runtime '. Please correct the usage

page 19, line 15 : change 'ADJ-based' to adjoint based

page 21, line 2 : either remove or replace 'observably' by significantly

---

## Author Comment (AC1) · 17 Sep 2019

Dear Reviewer,

Thank you for your helpful comments. Those comments are valuable for improving our paper, as well as the important guiding significance to our researches. The response to the comments are as following:

(1) Our paper aims to adopt a swarm intelligence algorithm based on the dimensional reduction strategy to solve the optimal precursors (OPRs) of the North Atlantic Oscillation (NAO) without the adjoint model. The solving performance is also one of the focus in this paper and has a certain significance in practical.

(2) In the studies of Jiang et al. (2013) and Dai et al. (2016), the algorithm they adopted to explore OPR is spectral projected gradient 2 (SPG2) [1]. The SPG2 was designed to solve the minimum problem of a nonlinear function under a set of constraints and has been widely used in the related research on CNOP [2-5]. The solving process using SPG2 needs to acquire the gradient information via the adjoint model. In their studies, they adopted the T21 quasigeostrophic (QG) global spectral model (T21L3), which is an ideal model with 3 layers, to simulate the NAO and selected the geopotential height as the variable to quantify the NAO. The adjoint model of the T21L3 is called to provide the gradient data for calculating extrema using SPG2. The description of the SPG2 is added to the section about related works in the revised version.

Our work is based on their studies, and we extend the CNOP method to the numerical model which does not have the adjoint model. We adopt a new generation of the fully-coupled model called the Community Earth System Model (CESM) to simulate the NAO, and CESM has no adjoint model. Thus, SPG2 and similar methods are not suitable for this situation. The proposed approach is adjoint-free and is optimized with multiple parallel frameworks. The reason we do not compare our results with the results of Jiang et al. and Dai et al. is no uniform standard to compare. It is almost meaningless to compare between different models (T21L3/CESM), different quantified variables (Geopotential height/Sea level pressure), different input (Potential vorticity/Zonal wind, Meridional wind, ...) and different resolution and solution space, etc.

(3) According to the definition, the OPR is the initial perturbation that is most likely to develop into a climate event under the constraint. The reasons for the climate events can be explained by OPR from a physical standpoint, and the nonlinear behaviors of the perturbations reveal the nonlinear effect of related factors to the climate event [6]. Finding the OPR that evolves into a climate event is helpful to understand the process

of the event onset [7]. It is possible to improve the forecast skill by detecting the initial perturbations within the sensitive area. Thus, it can provide information about the future development of the NAO, and the meaning of the OPRs will be explained in the revised version. Thank you for pointing this out.

(4) The main corrections and some explanation in equations:

Page 5, line 9: m denotes the number of spatial points, and n denotes the length of the time series.

Page 6, equation (6): $u\_0$ in this equation means any perturbation, and $u\_0^{*}(NAO^+)$ mentioned in line 6 of the same page denotes the OPR for $NAO^+$, which is a kind of initial CNOP (a special perturbation). The difference between them is the * symbol.

Page 6, equation (4): 'Trop' means 'tropical cyclone'. The equation (4) describes the objective function in the research of the identification of the sensitive area for the tropical cyclone, and the constraint chosen in this work adopts this function. We distinguish the objective function for tropical cyclone from the objective function of the NAO using the subscript 'Trop'. u', v', t', pi' denote the initial perturbations of zonal wind, meridional wind, temperature and surface geopotential respectively.

Page 6, line 20: The nonlinear model propagator denotes the integration process defined in the kinetic equations of the numerical model. Briefly, it stands for the nonlinear process of the model.

Page 7, equation (7): $S\_i$ denotes the $i^{th}$ sample data and n is the number of the samples. This equation handles the sample by subtracting the mean value of all samples and weighting according to the area of the grid. We have changed lat_i into lat(i), thank you.

Page 7, equation (8): The L is the eigenvector matrix, and Sigma is a diagonal matrix whose entries in the main diagonal are the corresponding eigenvalues.

Page 7, line 24: The random values of position (X) and the speed (V) obey the normal

distribution and ensure the perturbation satisfies the constraints given in equation (5).

Page 7, line 25: L_{1, ..., m} is an m times m eigenvector matrix, but is not a diagonal matrix (See the explanation for Page 7, equation (8)).

Page 4, line 18: "To explore the process of nonlinear..." means that we aim to find out what role the nonlinear played during the simulation. We're sorry for our vague expression.

Page 18, line 11-12: The sentence means that the offspring particles would be generated through crossover operation and mutation with probability, and if the offspring particle is generated, the objective function value (fitness value) would be calculated.

We are sorry for our poor English and expression, and we will try our best to improve the manuscript. Once again, thank you very much for your comments and suggestions.

* * *
[Figure]

**Supplement:**

[revised manuscript text omitted]

evidence that CNOP is a useful method to investigate the onset of the NAO event. In their studies, the T21L3 quasigeostrophic global spectral model, which is a simple three-level model designed by Marshall and Molteni, is applied under ideal conditions (Marshall and Molteni, 1993). Due to the feature of the T21L3 model, they adopted geopotential height as the characterized variable and selected potential vorticity as the input variable. For solving CNOP, they all used spectral projected gradient 2 (SPG2) algorithm (Birgin et al., 2001). The SPG2 was designed to solve the minimum problem with restraints by determining the gradients of the cost function (Guo-Dong, 2009). Several similar approaches have been also adopted to calculate the CNOP, such as the sequential quadratic programming (SQP) algorithm (Barclay et al., 1998) and the limited memory Broyden-Fletcher-Goldfarb-Shanno (L-BFGS) algorithm (Liu and Nocedal, 1989). These traditional methods of numerical optimal have been widely applied in the studies of the CNOP in the early years (Duan and Mu, 2006; Wang et al., 2012; Bo et al., 2014). Since these algorithms rely on gradient information, the corresponding adjoint model needs to be called to obtain the gradients of the initial condition in the solving process.

However, the traditional adjoint-based algorithms are not feasible to solve CNOP in complicated operational models that do not have an adjoint available (Wang, 2010). In addition, the past research suggests that the adjoint-based method would fail with large initial disturbance or long prediction time due to the strong nonlinearity of the dynamical model. The local CNOPs would be produced by the adjoint-based method with high probability when the objective function has multiple extreme values. In recent years, the swarm intelligence algorithms are gradually put forward to the research of the CNOP (Zheng et al., 2012; Yang et al., 2017). These algorithms determine the search directions and obtain the extremum by updating the position of the particles. Since the search process of these algorithms does not need any gradient information, they can be extended to the implementation of the CNOP using numerical models without the adjoint model. It is also indicated that the swarm intelligence method still achieves global CNOP and has a shorter run time in the situation of larger initial perturbations, longer prediction times, multiple extrema values (Zheng et al., 2017) and discontinuous objective functions (Mu et al., 2015). Although the above algorithms are effective, it is very time-consuming to calculate CNOP in the original space. To enhance the performance of solving CNOP with complex numeric models, the researchers proposed intelligent algorithms based on feature extraction. The algorithms transform the problems in original input space with high dimensions into the problems in low dimension space. At present, the tentative application of intelligent algorithms based on feature extraction in solving CNOP yielded considerable achievements. The principal component analysis based genetic algorithm (PCAGA) (Zhang et al., 2017a), the Modified Artificial Bee Colony Algorithm (MABC) (Ren et al., 2016), the dynamic search Fireworks Algorithm with linearly decreased dimension number strategy (ld-dynFWA) (Mu et al., 2017a) and PCA based Flower Pollination (PCAFP) (Yuan et al., 2016) have been successfully adopted in tropical cyclone adaptive observations, El Niño-Southern Oscillation and double-gyre variation. The CNOPs obtained by these methods have similar patterns and larger fitness values in comparison to the adjoint method. It is illustrated that the PCA-based intelligent algorithm is appropriate for high dimensional numerical models, especially the models without the adjoint model.

The objective of this paper is to find the OPRs which produce the NAO anomaly pattern and explore the effect of the nonlinear process. We study the case using the Community Earth System Model (CESM), which is an ocean-atmosphere coupled model without an adjoint model. Thus, traditional algorithms like SPG2 are inappropriate for this case. In this paper,

we select the particle swarm optimization (PSO) and genetic algorithm (GA) hybrid algorithm (PSO-GA), which is an effective swarm algorithm that has been previously proposed (Chang et al., 2013; Nik et al., 2016; Kumar and Vidyarthi, 2016; Agarwal and Srivastava, 2018), to solve CNOP for the NAO events. The parallel adjoint-free algorithm called PGAPSO is combined PSO-GA with the principal component analysis (PCA) strategy and is optimized with multiple frameworks. The OPRs obtained by the proposed algorithm steadily produce the SLP anomaly mode and trigger the high NAOI. Compared to the PCA-based PSO (PPSO), the algorithm is improved to avoid falling into the local optimum and accelerates convergence. After parallelized with MPI and CUDA, the speed-up ratio of the intelligent solution system reaches $40\times$ compared with its serial version.

The structure of this paper is organized as follows: Section 2 describes the CESM, and section 3 presents the CNOP method, the PGAPSO algorithm and the parallelization technique. Experiments and results are displayed in section 4. This paper ends with a conclusion and future work in section 5.

**2 Community Earth System Model**

The CESM (Kay et al., 2015) is a new generation of fully coupled climate models developed in 2010. It has been widely used to simulate the carbon cycle (Lehner et al., 2015), ocean currents (Large and Caron, 2015), soil moisture (Swenson and Lawrence, 2012), precipitation (Hagos et al., 2016) and other climate phenomena. As shown in Figure 1, the CESM is composed of seven geophysical model components, respectively Atmospheric (Community Atmosphere Model, CAM), Sea- ice (CICE), Land (Community Land Model, CLM), River-runoff (River Transport Model, RTM), Ocean (Parallel Ocean Program, POP), Land-ice (CISM), Ocean-wave (XWAV). The CESM also has a Coupler (CPL) that coordinates the time evolution of geophysical models and delivers information between these components.

[Figure]

**Figure 1.** Main components of CESM.

The atmospherical component in CESM 1.2.2 is used to simulate the NAO in this work. CAM version 5.3, which is a global atmospherical general circulation model developed from the NCAR CCM3, is released as the atmosphere component of CESM 1.2. The CAM incorporates an interactive aerosol model where aerosols interact with the tropospheric chemistry. The component set we selected is *F_2000* that includes CAM, CLM and CICE(prescribed mode) activated with SST data mode,

and the CAM component is stand-alone. The parameter *nhtfrq* is set to -24, which denotes the daily average. We perform the experiments on a $0.9°\times1.25°$ horizontal grid with 26 levels in the vertical. The region we focus on, which is also the NAO mainly located at, is a two-dimensional domain consisting of $65\times105$ grids with the North Atlantic area between $20°N$ and $80°N$ and between $90°W$ and $40°E$.

**3 CNOP and PGAPSO**

**3.1 CNOP**

The CNOP is a natural extension of the linear singular vector into the nonlinear regime, and is proposed to study predictability problems of weather and climate in numerical models (Mu et al., 2009). OPR is a kind of initial perturbation that can trigger the largest uncertainty in prediction, and it can be solved by the CNOP method. Specifically, the objective function achieves the maximum under the constraint condition at prediction time by superimposing OPRs on the basic state. In this experiment, we choose a blocking indicator proposed by Liu (Liu, 1994) to quantify the extent of the NAO events. The NAOI is defined as the projection of the SLP field on the NAO anomaly pattern:

$$NAOI = \frac{\langle SLP_{NAO}, SLP_d \rangle}{\langle SLP_{NAO}, SLP_{NAO} \rangle} \tag{1}$$

where $SLP_d$ is obtained by subtracting the climatological mean from SLP output, and $\langle \rangle$ denotes the inner product operation of vectors. $SLP_{NAO}$ denotes the NAO anomaly pattern acquired by the empirical orthogonal function (EOF) analysis. The EOF is a widely used tool to decompose the spatial-temporal distribution features in geonomy (Baldwin and Dunkerton, 1999). The procedures of EOF are listed as follows:

- Process the SLP historical data into anomaly values by subtracting the mean climate state of 10-year SLP time series data, recorded as $X_{m\times n}$. m denotes the number of spatial points, and n denotes the length of the time series.

- Calculate the covariance matrix $C_{m\times m}$ via: $C_{m\times m} = \frac{1}{n}X \times X^T$.

- Solve the eigenvalues ($\lambda_{1,...,m}$) and eigenvectors ($V_{m\times m}$) of $C_{m\times m}$ with the constraint condition: $C_{m\times m} \times V_{m\times m} = V_{m\times m} \times \Lambda_{m\times m}$.

- The eigenvectors corresponding to $\lambda_k$ is the $k^{th}$ column of $V_{m\times m}$, that is, $EOF_k = V(:,k)$.

In general, the first mode decomposed by EOF is chosen as the NAO anomaly pattern, which is illustrated in Figure 2. The NAO spatial pattern is manifested as a typical meridional dipole mode, which consists of the Iceland low pressure along with the North Atlantic subtropical high. In Figure 2, it is a positive phase of the NAO, presenting the mode with the negative anomalies in high latitude and the positive anomalies in low latitude.

[Figure]

**Figure 2.** The first mode of the EOF in SLP anomaly field concentrated in the North Atlantic region between $90°W$ - $40°E$, $20°N$ - $80°N$.

The procedure for solving CNOP can be regarded as the following extrema problem:

$$J(u_0^*)_{NAO^+} = \max_{\|u_0\| \leq \sigma} J(u_0) = NAOI(NAO^+)_{CNOP} - NAOI_{refer} \tag{2}$$

$$J(u_0^*)_{NAO^-} = \min_{\|u_0\| \leq \sigma} J(u_0) = NAOI(NAO^-)_{CNOP} - NAOI_{refer} \tag{3}$$

where $u_0$ is the vector of physics variables listed in Table 1. $J(u_0)$ is the objective function defined by the difference between
5    NAOI triggered by perturbation $u_0$ in the final state and NAOI in the reference state. In formula (2) and (3), the perturbation $u_0^*(NAO^+)$ makes $J(u_0)$ achieve the maximum, whereas $u_0^*(NAO^-)$ makes $J(u_0)$ achieve the minimum. According to the definition of the OPR, $u_0^*(NAO^+)$ is the OPR of the $NAO^+$, and $u_0^*(NAO^-)$ is the OPR of the $NAO^-$.

**Table 1.** The related variables included in the perturbations.

| Variable name | Description | Units |
|:---:|:---:|:---:|
| U | Zonal wind | $m/s$ |
| V | Meridional wind | $m/s$ |
| T | Temperature | $K$ |
| Q | Specific humidity | $kg/kg$ |
| PS | Surface pressure | $Pa$ |
| PHIS | Surface geopotential | $m^2/s^2$ |

$\sigma$ denotes the constraint condition of the OPRs. The constraint condition we used in this paper is consulted from a similar study in the field of the atmosphere. In the study of the identification of the sensitive areas for tropical cyclone using CNOP,

the summation of kinetic energy is chosen as the objective function, available relative potential and surface potential energy in the verification areas $D$ (Zhang et al., 2017a):

$$J(u_0)_{Trop} = \frac{1}{D} \int_D \int_0^1 [u'^2 + v'^2 + \frac{C_p}{T_r}t'^2 + R_a T_r (\frac{\pi'}{\pi_r})^2]d\sigma dD \tag{4}$$

where $J(u_0)_{Trop}$ is the objective function in the research of tropical cyclone. $u'$, $v'$, $t'$ and $\pi'$ are the initial perturbations of zonal wind, meridional wind, temperature and surface geopotential respectively. $C_p$ is the specific heat at the constant pressure which is set to 1005.7 $J \cdot kg^{-1}K^{-1}$ and $T_r$ is the reference temperature with a value of $270K$. $R_a$ denotes the ideal gas constant, and its value is set to 287.05 $J \cdot kg^{-1}K^{-1}$. $\pi_r$ is the reference static pressure with a value of 1000 $hPa$. In order to ensure the perturbations within a reasonable range, the constraint is set to 10% of the dry energy norm in the basic state, that is:

$$\sigma = 10\% * \frac{1}{D} \int_D \int_0^1 [U_0{}^2 + V_0{}^2 + \frac{C_p}{T_r}T_0{}^2 + R_a T_r (\frac{\Pi}{\pi_r})^2]d\sigma dD \tag{5}$$

We adopt the above constraint $\sigma$ since our work and the research of tropical cyclone have the same variables. The constraint is to ensure the reasonability of these variables and avoid the appearance of abnormal values. The feasibility of restraining these variables using this constraint has been proved in (Zhou and Mu, 2011; Zhang et al., 2017a, 2018). Combining the formula (1), (2) and (3), the objective function is described as follows:

$$\begin{aligned} J(u_0) &= \Delta NAOI \\ &= \frac{\langle M_{t_0 \to T}(U_0 + u_0) - M_{t_0 \to T}(U_0), SLP_{NAO} \rangle}{\langle SLP_{NAO}, SLP_{NAO} \rangle} \end{aligned} \tag{6}$$

where $M_{t_0 \to T}$ represents the nonlinear propagator that "propagates" the initial state in time $t_0$ to the prediction time $T$, and $U_0$ denotes the initial basic state. Therefore, $M_{t_0 \to T}(U_0)$ denotes the reference state at prediction time $T$. The objective function is the projection of the SLP field difference between the final state and the reference state on the NAO anomaly pattern.

**3.2 PGAPSO**

Under the resolution of *f09_g16* with an approximate grid spacing of $0.9° \times 1.25°$, the total dimensions of variables involved in the objective function are 5861376. It is difficult for the algorithm to solve the optimization problem in such high dimensions. Thus, we need to extract the feature of samples to reduce the data scale.

PCA is a traditional method for feature extraction and has been widely used in signal separation (Kasban et al., 2016), environment forecasting (ULSAUFIE et al., 2013) and pattern classification (Li et al., 2017), etc. In this paper, we adopt PCA to implement dimension reduction for sample data. After running 10-year integration (only in winter) on a daily average using

CESM, we extract the variables displayed in Table 1 from the model files. The size of $U$, $V$, $T$ and $Q$ is 26 (layer) $\times$ 192 (latitude) $\times$ 288 (longitude), and the size of $PS$ and $PHIS$ is 192 $\times$ 288. Each piece of sample is handled into a vector with one dimension (1 $\times$ 5861376), and the original sample is a matrix of 900 $\times$ 5861376 (containing 900 days). Then subtract the climatological mean of the 10-year data from each sample, and the obtained sample is weighted according to the area of the grid:

$$S_i = (S_i - \frac{1}{n} \sum S_i) * \cos(lat(i)) \ (i = 1, 2, \ldots, n) \tag{7}$$

where $S_i$ denotes the $i^{th}$ sample data and $n$ is the number of the samples. $lat(i)$ is the latitude of the $i^{th}$ row in the grid, and the weight is calculated approximately via the cosine value of the grid's latitude. Then the eigenvalues $(\lambda_1, \ldots, \lambda_n)$ and eigenvectors of the covariance matrix $SS^T$ are calculated to obtain principal components:

$$SS^T L = L\Sigma \tag{8}$$

where $L$ is the eigenvector matrix, and $\Sigma$ is a diagonal matrix whose entries in the main diagonal are the corresponding eigenvalues. The top $m$ columns of the eigenvectors $L$ sorting by their eigenvalues are selected as the principal components. The value of $m$ is determined by the contribution rate, which is defined as:

$$r = \frac{\sum_{i=1}^{m} \lambda_i}{\sum_{i=1}^{n} \lambda_i} \tag{9}$$

In this work, $m$ is set to the minimum number of columns that meet the contribution rate of 95%. The reduced space with $m$ dimensions is far smaller than the original one.

To obtain the extremum of the objective function, we adopt a hybrid algorithm improved from two classical algorithms, PSO and GA. The PSO is a type of intelligent heuristic algorithm to solve the problem with NP property (Kennedy, 2011). The position with the best fitness value is searched by tracing individual optimal positions and the optimal global position in the meantime. The flow of the algorithm is described in brief: (1) Initialize the speed ($V$) and position ($X$) of particle swarm with random values. The random values obey the normal distribution and ensure the perturbations satisfy the constraints. (2) For each particle $i$, the position vectors in reduced space need to be restored into original space via $X_i' = X_i \cdot L_{1,\ldots,m}$, thereinto, $L$ is an m times m eigenvector matrix. Then superpose the perturbation $X_i'$ on the basic state. When the model integration is finished, calculate the fitness value of each particle through the formula (6) and record its optimal position ($X_{pb}$) along with the global optimal position ($X_{gb}$). (3) Update the position and speed of each particle. The updating formula is as follows:

$$\begin{cases} V_i^{k+1} = \omega_k V_i^k + c_1 r_1 (X_{pb}^k - X_i^k) + c_2 r_2 (X_{gb}^k - X_i^k) \\ X_i^{k+1} = X_i^k + V_i^{k+1} \end{cases} \tag{10}$$

where $V_i^k$ is the speed of particle $i$ for step $k$ and $V_i^{k+1}$ is for step $k+1$. $c_1$ is the self-awareness coefficient for the historical self-optimal position and $c_2$ is the social-awareness coefficient for global optimal position of all particles. The empirical value of $c_1$ and $c_2$ are both set to 2. $r_1$ and $r_2$ are random float numbers with uniform distribution in $[0,1]$. $X_{pb}^k$ refers to the position of particle $i$ where objective function acquires the maximum(minimum) in $k$ steps, and $X_{gb}^k$ represents the position where objective function achieves global extrema in $k$ turns. Both position vectors and speed vectors are in reduced space with $m$ dimensions. $\omega_k$ is the weight parameter and calculated by:

$$\omega_k = \omega_{max} - \frac{\omega_{max} - \omega_{min}}{iter_{max}} * iter \tag{11}$$

where $iter$ is the current number of step, and $iter_{max}$ is set to 100.

In PGAPSO, PSO is viewed as the main body of the search process, and the GA further optimizes the position. As a meta-heuristic algorithm, the GA derives from natural selection (Goldberg and Holland, 1988). When the fitness value is obtained in step (2) of PSO, the particles are fed into GA for further search. A portion of particles in the existing population are selected according to their fitness value to breed a new generation. The selection operation is performed on roulette strategy, that is to say, the probability that each individual is selected is equal to the ratio of its fitness value to the total fitness value of the entire population:

$$p_s = \frac{J(u_{X_i'})}{\Sigma J(u_{X'})} \tag{12}$$

After that, the selected parents generate new individuals via crossover:

$$X_a'\{x_s,\ldots,x_e\} = X_b\{x_s,\ldots,x_e\}$$
$$X_b'\{x_s,\ldots,x_e\} = X_a\{x_s,\ldots,x_e\} \tag{13}$$

Then the new generation mutates with probability $p_m$ in a single position to avoid genetic drift. The fitness value of each new generation is compared against its parents, and the best position is recorded. If the new individual has a better fitness value compared with the global best position, the global best position ($X_{gb}^k$) would be replaced by the new position. With the optimal local position and global optimal position, the speed and position of particles are updated using formula (10). The final global fitness value is obtained until the $iter$ reaches $iter_{max}$ or the norm of particles' speed reaches the specific threshold.

**3.3 Parallelization**

The computation of CNOP in CESM is quite time-consuming. With 48 CPU cores, 30 particles and 100 iterations, it takes about 13.75 days to obtain the OPRs in the serial program. For PGAPSO to operate more effectively, multiple parallel techniques and frameworks are adopted in this work.

**3.3.1 CESM Parallelization**

The role of the CAM component in CESM is to simulate the variation of atmosphere, and the largest variation can be discovered by objective function using PGAPSO. With high resolution, the input data handled for integration in nonlinear processes of CAM possess the features as massive variables, high dimensions and complexity, which makes the invocation for CAM become the primary time-consuming task in the whole program. Although CESM has already been parallelized using Message Passing Interface (MPI) and Open Multi-Processing (OpenMP), it is still time-consuming.

Recently, the Graphics Processing Unit (GPU) has been widely used in accelerating numerical models. Since GPU is suitable for parallel computing on a large scale, it can significantly improve the execution performance of climate models. A parallel scheme for Community Climate System Model (CCSM) has been proposed to shorten the runtime of climate prediction by porting the radiation module onto GPUs (Coleman and Feldman, 2013). The module was parallelized using the inline method and communicated with MPI routines. A cloud analysis scheme called Goddard Cumulus Ensemble (GCE) in Weather Research and Forecasting (WRF) was highly expedited using NVIDIA Tesla K40 with 2880 cores (Huang et al., 2015). Compared to the CPU-based parallel version running on 4 nodes, the GPU-based scheme performed faster. As for CESM, the novel asynchronous execution strategy has provided significant performance benefits (Korwar et al., 2013). The most time-consuming routines have been accelerated via OpenACC directives and achieved a speedup of $1.19\times$-$1.53\times$ for the entire model. Another attempt for accelerating CESM was to port CESM along with a rewritten vertical remapping scheme onto GPUs (Carpenter et al., 2013). The results indicated that the performance of the optimized subroutine was improved substantially. Related works show that GPU is an alternative approach to enhance the performance of the climate model.

In this work, we port several time-consuming subroutines in CAM onto GPUs through the PGI CUDA Fortran interface. After analysis run time using *pref*, shown in Figure 3, subroutine *radclwmx* and *radabs* both consume longer runtime compared with other subroutines. These two subroutines are both optimized with the CUDA platform. Simultaneously, kernel directives and OpenACC directives are used to implement the simplification of specific operations on the device. The function execution and data replication are overlapped using asynchronous streams. 
[revised manuscript text omitted]

The trends for these three states at another start date are shown in Figure 6. This figure also illustrates for 5-day optimization, 7-day optimization and 15-day optimization. In Figure 6, similar to Figure 5, the NAOI triggered by CNOP always has a big gap with the reference flow and achieves an abnormal high value in the final period. Similarly, the flows of $NAO^+$ and $NAO^-$ both noticeably deviate from the reference flow in the last few days.

[Figure]

**Figure 6.** Same as Figure 5, but for another start date.

To evaluate the NAOI of CNOPs more visually, Table 3 reports the incremental values of the NAOI with different simulation time in Figure 5 and Figure 6. From Table 3, the difference between the NAOI in the final state and the NAOI in reference state increases when the integration time becomes longer. We can also find that the result depends on the start date. Although large discrepancy exists between the $\Delta NAOI$ with a simulation time of 15 days in Figure 5 and Figure 6, the algorithm can always find the CNOPs that can cause the abnormal state, and $|\Delta NAOI|$ is far greater than 1.

**Table 3.** The increment value of NAOI with different simulation time in Figure 5 and Figure 6.

[revised manuscript text omitted]

5 ### 4.3 Performance Analysis

In order to demonstrate the performance improvement of parallel PGAPSO adopted in this paper, Figure 10 compares the runtime of parallel PGAPSO and serial PGAPSO for one iteration. The runtime of CESM is the performance bottleneck of the algorithm, which can be broken by running in parallel. Our parallel scheme using MPI implements the simultaneous execution of multiple particles to solve the problem. From Figure 10, we can see that when the number of CPU cores is more than 840, it
10 will take longer to run the serial algorithm. CESM has been paralleled with MPI and OpenMP; when the number of CPU cores increases to the critical point, the frequent communication would make the runtime of the CESM increase. The speedup ratio of parallel PGAPSO compared with serial PGAPSO is displayed in Table 4. The speedup ratio increases with the rise of the CPU cores' number. With assigning CPU cores to multiple tasks, the execution time of parallel PGAPSO continues to decline, while the serial PGAPSO takes longer owing to communication. With 1080 CPU cores, PGAPSO based on the parallel scheme
15 achieves a speedup of 40× compared to its serial version.

**Table 4.** The speedup of parallel PGAPSO compared with serial PGAPSO.

[revised manuscript text omitted]

Zhang, X., Mu, M., Wang, Q., and Pierini, S.: Optimal precursors triggering the Kuroshio Extension state transition obtained by the Condi-

20  tional Nonlinear Optimal Perturbation approach, Advances in Atmospheric Sciences, 34, 685–699, 2017b.

Zheng, Q., Dai, Y., Zhang, L., Sha, J., and Xiaoqing, L. U.: On the Application of a Genetic Algorithm to the Predictability Problems Involving "On-Off" Switches, Advances in Atmospheric Sciences, 29, 422–434, 2012.

Zheng, Q., Yang, Z., Sha, J., and Yan, J.: Conditional nonlinear optimal perturbations based on the particle swarm optimization and their applications to the predictability problems, Nonlinear Processes in Geophysics, 24, 101–112, 2017.

25  Zhou, F. and Mu, M. U.: The Impact of Verification Area Design on Tropical Cyclone Targeted Observations Based on the CNOP Method, Advances in Atmospheric Sciences, 28, 997, 2011.

---

## Author Comment (AC2) · 17 Sep 2019

Dear Editor,

Thank you very much for your careful review and constructive suggestions concerning our manuscript. These comments are very valuable and helpful for us to revise and improve our paper. We have studied comments carefully and have made corrections. The response to the comments are as follows:

1. The optimization procedure of PSOGA has indeed been previously proposed [1-

[Figure]

4]. Our work combines it with principal component analysis (PCA) strategy, and the algorithm is applied in the research of the OPR for the NAO using the CNOP approach for the first time. The related references are attached according to your suggestion.

2. When the individuals in PSO are being mutated or crossed-over in the GA phase, the new individuals are generated with new positions. If these new positions have a better fitness value compared with the global best position, the global best position (X_gb) would be replaced by the new position. Then the population is updated.

3. As the common practice in previous works [5-8], a sample with multiple variables is reshaped into a vector with one dimension. For instance, the scale of U, V, T and Q is 26 * 192 * 288, and the scale of PS and PHIS is 192 * 288. Then one piece of sample is handled into a 1 (row) times 5861376 (column) vector, and the different variables are not weighted. We have 900 pieces of samples, we conduct PCA on the 900 (row) times 5861376 (column) matrix. Therefore, we can not separate these six variables from the output of the PCA, and we do not need to. Before the perturbations are superimposed on the input of the CESM, the position would be restored into the original dimensions via inverse transform.

4. Yes, the step corresponding to "compare parents and offspring" requires invoking CESM, and we have made corrections in the flow chart. Thank you for pointing it out.

5. The notation is corrected. Thank you.

6. Since the reference state is the result without perturbations, the reference flows displayed in the three panels are the same. We have added the description. The initial reference state is randomly selected from a winter day in the model year. Based on the series of experimental results, the NAOI value is related to the initial reference state, but the proposed algorithm can always obtain the optimal for both NAOˆ+ and NAOˆ- and find out the OPRs that can trigger the NAO events. To prove the results have universality, we have added an experiment with a different initial reference state, and illustrate the trend of the NAOI in perturbation states. To avoid repetitive analysis, the

pattern and nonlinear evolution are still illustrated according to the scenario in Figure 5.

7. The previous studies indicate that the NAO events can be viewed as a stochastic process with an intrinsic time scale of 2 weeks [9]. Very recently, the research result of our co-author in this paper suggested that the skillful forecast time of the NAO is about 2 weeks [10]. Longer simulation may contain multiple variation processes. Thus, we select the optimization time within 15 days.

8. The random perturbations are completely random in the search process. Due to the limitation of the chart, we can only display the general trend that most of the perturbations show. We want to explain that there are significant differences between the CNOPs obtained from the proposed algorithm and the random perturbations. It may be an inappropriate way to analysis using several random perturbations, and we have deleted this part from the section.

9. The parallel scheme can only enhance the efficiency and shorten the runtime of the calculation process. Since the parallel strategy just makes the multiple particles search concurrently, it would not influence the result of NAOI. With the same number of searches, the parallel OPR would have a similar structure and play an equal role compared with the serial OPR in theory.

10. The perturbations are restored into original space via multiplying the particle position and the components acquired from PCA, and the particle position is adjusted from the initial random position. Therefore, the perturbation field depends on the random seed. The range of the random position is chosen to ensure most of the perturbations in the original space obey the constraint. Besides, the random range should make the perturbation achieve better fitness value as possible. The random range is the empirical value obtained from multiple experiments and can affect the result of the algorithm.

[1] Chang J X , Bai T , Huang Q , et al. Optimization of Water Resources Utilization by PSO-GA[J]. Water Resources Management, 2013, 27(10):3525-3540.
[2] Nik A A , Nejad F M , Zakeri H . Hybrid PSO and GA approach for optimizing surveyed asphalt pavement inspection units in massive network[J]. Automation in Construction, 2016:S0926580516301571.

[3] Kumar N , Vidyarthi D P . A novel hybrid PSO–GA meta-heuristic for scheduling of DAG with communication on multiprocessor systems[J]. Engineering with Computers, 2016, 32(1):35-47.

[4] Agarwal M, Srivastava G M S. Genetic Algorithm-Enabled Particle Swarm Optimization (PSOGA)-Based Task Scheduling in Cloud Computing Environment[J]. International Journal of Information Technology & Decision Making, 2018, 17(04): 1237-1267.

[5] Yuan S, Li M, Mu B, et al. PCAFP for solving CNOP in double-gyre variation and its parallelization on clusters[C]//2016 IEEE 18th International Conference on High Performance Computing and Communications; IEEE 14th International Conference on Smart City; IEEE 2nd International Conference on Data Science and Systems (HPCC/SmartCity/DSS). IEEE, 2016: 284-291.

[6] Mu B, Zhao J, Yuan S, et al. Parallel dynamic search fireworks algorithm with linearly decreased dimension number strategy for solving conditional nonlinear optimal perturbation[C]//2017 International Joint Conference on Neural Networks (IJCNN). IEEE, 2017: 2314-2321.

[7] Zhang L L, Yuan S J, Mu B, et al. CNOP-based sensitive areas identification for tropical cyclone adaptive observations with PCAGA method[J]. Asia-Pacific Journal of Atmospheric Sciences, 2017, 53(1): 63-73.

[8] Mu B, Ren J, Yuan S, et al. Identifying Typhoon Targeted Observations Sensitive Areas Using the Gradient Definition Based Method[J]. Asia-Pacific Journal of Atmospheric Sciences, 2019, 55(2): 195-207.

[9] Feldstein S B . The dynamics of NAO teleconnection pattern growth and decay[J]. Quarterly Journal of the Royal Meteorological Society, 2003, 129(589):901-924.

[10] Dai, G. K„ M. Mu, and Z. M. Jiang, 2019: Evaluation of the forecast performance for North Atlantic Oscillation onset. Adv. Atmos. Sci., 36(7), 000–000, https://doi.org/10.1007/s00376-019-8277-9.

Please also note the supplement to this comment:
https://www.nonlin-processes-geophys-discuss.net/npg-2019-25/npg-2019-25-AC2-supplement.pdf
* * *
[Figure]

**Supplement:**

[revised manuscript text omitted]

evidence that CNOP is a useful method to investigate the onset of the NAO event. In their studies, the T21L3 quasigeostrophic global spectral model, which is a simple three-level model designed by Marshall and Molteni, is applied under ideal conditions (Marshall and Molteni, 1993). Due to the feature of the T21L3 model, they adopted geopotential height as the characterized variable and selected potential vorticity as the input variable. For solving CNOP, they all used spectral projected gradient 2

- 5 (SPG2) algorithm (Birgin et al., 2001). The SPG2 was designed to solve the minimum problem with restraints by determining the gradients of the cost function (Guo-Dong, 2009). Several similar approaches have been also adopted to calculate the CNOP, such as the sequential quadratic programming (SQP) algorithm (Barclay et al., 1998) and the limited memory Broyden-Fletcher-Goldfarb-Shanno (L-BFGS) algorithm (Liu and Nocedal, 1989). These traditional methods of numerical optimal have been widely applied in the studies of the CNOP in the early years (Duan and Mu, 2006; Wang et al., 2012; Bo et al., 2014).
- 10 Since these algorithms rely on gradient information, the corresponding adjoint model needs to be called to obtain the gradients of the initial condition in the solving process.

However, the traditional adjoint-based algorithms are not feasible to solve CNOP in complicated operational models that do not have an adjoint available (Wang, 2010). In addition, the past research suggests that the adjoint-based method would fail with large initial disturbance or long prediction time due to the strong nonlinearity of the dynamical model. The local CNOPs

- 15 would be produced by the adjoint-based method with high probability when the objective function has multiple extreme values. In recent years, the swarm intelligence algorithms are gradually put forward to the research of the CNOP (Zheng et al., 2012; Yang et al., 2017). These algorithms determine the search directions and obtain the extremum by updating the position of the particles. Since the search process of these algorithms does not need any gradient information, they can be extended to the implementation of the CNOP using numerical models without the adjoint model. It is also indicated that the swarm
- 20 intelligence method still achieves global CNOP and has a shorter run time in the situation of larger initial perturbations, longer prediction times, multiple extrema values (Zheng et al., 2017) and discontinuous objective functions (Mu et al., 2015). Although the above algorithms are effective, it is very time-consuming to calculate CNOP in the original space. To enhance the performance of solving CNOP with complex numeric models, the researchers proposed intelligent algorithms based on feature extraction. The algorithms transform the problems in original input space with high dimensions into the problems in
- 25 low dimension space. At present, the tentative application of intelligent algorithms based on feature extraction in solving CNOP yielded considerable achievements. The principal component analysis based genetic algorithm (PCAGA) (Zhang et al., 2017a), the Modified Artificial Bee Colony Algorithm (MABC) (Ren et al., 2016), the dynamic search Fireworks Algorithm with linearly decreased dimension number strategy (ld-dynFWA) (Mu et al., 2017a) and PCA based Flower Pollination (PCAFP) (Yuan et al., 2016) have been successfully adopted in tropical cyclone adaptive observations, El Niño-Southern Oscillation and
- 30 double-gyre variation. The CNOPs obtained by these methods have similar patterns and larger fitness values in comparison to the adjoint method. It is illustrated that the PCA-based intelligent algorithm is appropriate for high dimensional numerical models, especially the models without the adjoint model.

The objective of this paper is to find the OPRs which produce the NAO anomaly pattern and explore the effect of the nonlinear process. We study the case using the Community Earth System Model (CESM), which is an ocean-atmosphere

35 coupled model without an adjoint model. Thus, traditional algorithms like SPG2 are inappropriate for this case. In this paper,

we select the particle swarm optimization (PSO) and genetic algorithm (GA) hybrid algorithm (PSO-GA), which is an effective swarm algorithm that has been previously proposed (Chang et al., 2013; Nik et al., 2016; Kumar and Vidyarthi, 2016; Agarwal and Srivastava, 2018), to solve CNOP for the NAO events. The parallel adjoint-free algorithm called PGAPSO is combined PSO-GA with the principal component analysis (PCA) strategy and is optimized with multiple frameworks. The OPRs obtained

5 by the proposed algorithm steadily produce the SLP anomaly mode and trigger the high NAOI. Compared to the PCA-based PSO (PPSO), the algorithm is improved to avoid falling into the local optimum and accelerates convergence. After parallelized with MPI and CUDA, the speed-up ratio of the intelligent solution system reaches 40× compared with its serial version.

The structure of this paper is organized as follows: Section 2 describes the CESM, and section 3 presents the CNOP method, the PGAPSO algorithm and the parallelization technique. Experiments and results are displayed in section 4. This paper ends

10 with a conclusion and future work in section 5.

**2 Community Earth System Model**

The CESM (Kay et al., 2015) is a new generation of fully coupled climate models developed in 2010. It has been widely used to simulate the carbon cycle (Lehner et al., 2015), ocean currents (Large and Caron, 2015), soil moisture (Swenson and Lawrence, 2012), precipitation (Hagos et al., 2016) and other climate phenomena. As shown in Figure 1, the CESM is composed of seven
geophysical model components, respectively Atmospheric (Community Atmosphere Model, CAM), Sea- ice (CICE), Land (Community Land Model, CLM), River-runoff (River Transport Model, RTM), Ocean (Parallel Ocean Program, POP), Land-ice (CISM), Ocean-wave (XWAV). The CESM also has a Coupler (CPL) that coordinates the time evolution of geophysical models and delivers information between these components.

Figure 1. Main components of CESM.

The atmospherical component in CESM 1.2.2 is used to simulate the NAO in this work. CAM version 5.3, which is a
global atmospherical general circulation model developed from the NCAR CCM3, is released as the atmosphere component of CESM 1.2. The CAM incorporates an interactive aerosol model where aerosols interact with the tropospheric chemistry. The component set we selected is *F*\_2000 that includes CAM, CLM and CICE(prescribed mode) activated with SST data mode,

and the CAM component is stand-alone. The parameter *nhtfrq* is set to -24, which denotes the daily average. We perform the experiments on a  $0.9^{\circ} \times 1.25^{\circ}$  horizontal grid with 26 levels in the vertical. The region we focus on, which is also the NAO mainly located at, is a two-dimensional domain consisting of  $65 \times 105$  grids with the North Atlantic area between  $20^{\circ}N$  and  $80^{\circ}N$  and between  $90^{\circ}W$  and  $40^{\circ}E$ .

**5 3 CNOP and PGAPSO**

**3.1 CNOP**

The CNOP is a natural extension of the linear singular vector into the nonlinear regime, and is proposed to study predictability problems of weather and climate in numerical models (Mu et al., 2009). OPR is a kind of initial perturbation that can trigger the largest uncertainty in prediction, and it can be solved by the CNOP method. Specifically, the objective function achieves the maximum under the constraint condition at prediction time by superimposing OPPs on the basis state. In this constraints

10 the maximum under the constraint condition at prediction time by superimposing OPRs on the basic state. In this experiment, we choose a blocking indicator proposed by Liu (Liu, 1994) to quantify the extent of the NAO events. The NAOI is defined as the projection of the SLP field on the NAO anomaly pattern:

$$NAOI = \frac{\langle SLP_{NAO}, SLP_d \rangle}{\langle SLP_{NAO}, SLP_{NAO} \rangle} \tag{1}$$

where  $SLP_d$  is obtained by subtracting the climatological mean from SLP output, and  $\langle \rangle$  denotes the inner product operation of 15 vectors.  $SLP_{NAO}$  denotes the NAO anomaly pattern acquired by the empirical orthogonal function (EOF) analysis. The EOF is a widely used tool to decompose the spatial-temporal distribution features in geonomy (Baldwin and Dunkerton, 1999). The procedures of EOF are listed as follows:

- Process the SLP historical data into anomaly values by subtracting the mean climate state of 10-year SLP time series data, recorded as  $X_{m \times n}$ . m denotes the number of spatial points, and n denotes the length of the time series.
- **20** Calculate the covariance matrix  $C_{m \times m}$  via:  $C_{m \times m} = \frac{1}{n} X \times X^T$ .
  - Solve the eigenvalues  $(\lambda_{1,...,m})$  and eigenvectors  $(V_{m \times m})$  of  $C_{m \times m}$  with the constraint condition:  $C_{m \times m} \times V_{m \times m} = V_{m \times m} \times \Lambda_{m \times m}$ .
  - The eigenvectors corresponding to  $\lambda_k$  is the  $k^{th}$  column of  $V_{m \times m}$ , that is,  $EOF_k = V(:,k)$ .

In general, the first mode decomposed by EOF is chosen as the NAO anomaly pattern, which is illustrated in Figure 2. The NAO spatial pattern is manifested as a typical meridional dipole mode, which consists of the Iceland low pressure along with the North Atlantic subtropical high. In Figure 2, it is a positive phase of the NAO, presenting the mode with the negative anomalies in high latitude and the positive anomalies in low latitude.

Figure 2. The first mode of the EOF in SLP anomaly field concentrated in the North Atlantic region between  $90^{\circ}W - 40^{\circ}E$ ,  $20^{\circ}N - 80^{\circ}N$ .

The procedure for solving CNOP can be regarded as the following extrema problem:

$$J(u_0^*)_{NAO^+} = \max_{\|u_0\| \le \sigma} J(u_0) = NAOI(NAO^+)_{CNOP} - NAOI_{refer}$$
(2)

$$J(u_0^*)_{NAO^-} = \min_{\|u_0\| \le \sigma} J(u_0) = NAOI(NAO^-)_{CNOP} - NAOI_{refer}$$
(3)

where  $u_0$  is the vector of physics variables listed in Table 1.  $J(u_0)$  is the objective function defined by the difference between 5 NAOI triggered by perturbation  $u_0$  in the final state and NAOI in the reference state. In formula (2) and (3), the perturbation  $u_0^*(NAO^+)$  makes  $J(u_0)$  achieve the maximum, whereas  $u_0^*(NAO^-)$  makes  $J(u_0)$  achieve the minimum. According to the definition of the OPR,  $u_0^*(NAO^+)$  is the OPR of the  $NAO^+$ , and  $u_0^*(NAO^-)$  is the OPR of the  $NAO^-$ .

Table 1. The related variables included in the perturbations.

| Variable name | Description          | Units     |
|---------------|----------------------|-----------|
| U             | Zonal wind           | m/s       |
| V             | Meridional wind      | m/s       |
| Т             | Temperature          | K         |
| Q             | Specific humidity    | kg/kg     |
| PS            | Surface pressure     | Pa        |
| PHIS          | Surface geopotential | $m^2/s^2$ |

 $\sigma$  denotes the constraint condition of the OPRs. The constraint condition we used in this paper is consulted from a similar study in the field of the atmosphere. In the study of the identification of the sensitive areas for tropical cyclone using CNOP,

the summation of kinetic energy is chosen as the objective function, available relative potential and surface potential energy in the verification areas D (Zhang et al., 2017a):

$$J(u_0)_{Trop} = \frac{1}{D} \int_{D} \int_{0}^{1} [u'^2 + v'^2 + \frac{C_p}{T_r} t'^2 + R_a T_r (\frac{\pi'}{\pi_r})^2] d\sigma dD$$
(4)

where  $J(u_0)_{Trop}$  is the objective function in the research of tropical cyclone. u', v', t' and  $\pi'$  are the initial perturbations of zonal wind, meridional wind, temperature and surface geopotential respectively.  $C_p$  is the specific heat at the constant pressure which is set to 1005.7  $J \cdot kg^{-1}K^{-1}$  and  $T_r$  is the reference temperature with a value of 270K.  $R_a$  denotes the ideal gas constant, and its value is set to 287.05  $J \cdot kg^{-1}K^{-1}$ .  $\pi_r$  is the reference static pressure with a value of 1000 hPa. In order to ensure the perturbations within a reasonable range, the constraint is set to 10% of the dry energy norm in the basic state, that is:

10
$$\sigma = 10\% * \frac{1}{D} \int_{D} \int_{0}^{1} [U_0^2 + V_0^2 + \frac{C_p}{T_r} T_0^2 + R_a T_r (\frac{\Pi}{\pi_r})^2] d\sigma dD$$
 (5)

We adopt the above constraint  $\sigma$  since our work and the research of tropical cyclone have the same variables. The constraint is to ensure the reasonability of these variables and avoid the appearance of abnormal values. The feasibility of restraining these variables using this constraint has been proved in (Zhou and Mu, 2011; Zhang et al., 2017a, 2018). Combining the formula (1), (2) and (3), the objective function is described as follows:

$$J(u_0) = \Delta NAOI$$

$$= \frac{\langle M_{t_0 \to T}(U_0 + u_0) - M_{t_0 \to T}(U_0), SLP_{NAO} \rangle}{\langle SLP_{NAO}, SLP_{NAO} \rangle}$$
(6)

where  $M_{t_0 \to T}$  represents the nonlinear propagator that "propagates" the initial state in time  $t_0$  to the prediction time T, and  $U_0$  denotes the initial basic state. Therefore,  $M_{t_0 \to T}(U_0)$  denotes the reference state at prediction time T. The objective function is the projection of the SLP field difference between the final state and the reference state on the NAO anomaly pattern.

**3.2 PGAPSO**

20 Under the resolution of  $f09\_g16$  with an approximate grid spacing of  $0.9^{\circ} \times 1.25^{\circ}$ , the total dimensions of variables involved in the objective function are 5861376. It is difficult for the algorithm to solve the optimization problem in such high dimensions. Thus, we need to extract the feature of samples to reduce the data scale.

PCA is a traditional method for feature extraction and has been widely used in signal separation (Kasban et al., 2016), environment forecasting (ULSAUFIE et al., 2013) and pattern classification (Li et al., 2017), etc. In this paper, we adopt PCA

25 to implement dimension reduction for sample data. After running 10-year integration (only in winter) on a daily average using

CESM, we extract the variables displayed in Table 1 from the model files. The size of U, V, T and Q is 26 (layer) × 192 (latitude) × 288 (longitude), and the size of PS and PHIS is 192 × 288. Each piece of sample is handled into a vector with one dimension (1 × 5861376), and the original sample is a matrix of 900 × 5861376 (containing 900 days). Then subtract the climatological mean of the 10-year data from each sample, and the obtained sample is weighted according to the area of the grid:

$$S_i = (S_i - \frac{1}{n} \sum S_i) * \cos(lat(i)) \ (i = 1, 2, \dots, n)$$
(7)

where  $S_i$  denotes the  $i^{th}$  sample data and n is the number of the samples. lat(i) is the latitude of the  $i^{th}$  row in the grid, and the weight is calculated approximately via the cosine value of the grid's latitude. Then the eigenvalues  $(\lambda_1, \ldots, \lambda_n)$  and eigenvectors of the covariance matrix  $SS^T$  are calculated to obtain principal components:

$$\quad SS^T L = L\Sigma \tag{8}$$

where L is the eigenvector matrix, and  $\Sigma$  is a diagonal matrix whose entries in the main diagonal are the corresponding eigenvalues. The top m columns of the eigenvectors L sorting by their eigenvalues are selected as the principal components. The value of m is determined by the contribution rate, which is defined as:

$$r = \frac{\sum_{i=1}^{m} \lambda_i}{\sum_{i=1}^{n} \lambda_i} \tag{9}$$

[revised manuscript text omitted]

---

## Author Comment (AC3) · 17 Sep 2019

Dear Reviewer,

Thank you very much for your attention and the comments on our paper. We have studied these comments carefully and tried our best to revise the manuscript. Responds to the comments are listed as follows:

1. In previous studies, the geopotential height is used to quantify the NAO due to the feature of the T21 quasigeostrophic (QG) global spectral (T21L3) model. In our

[Figure]

work, with a more complex physical process in the ocean-atmosphere coupled model, sea level pressure (SLP) is selected as the characterized variable of the North Atlantic Oscillation (NAO). The sentence on page 3, line 1-2 is not very accurate, and we would rewrite this part and make it well articulated.

2. In this paper, we propose a parallel hybrid algorithm to investigate the optimal precursor (OPR) of the NAO using conditional nonlinear optimal perturbation (CNOP) method. In the related works, the algorithm for studying the OPR of the NAO is based on the gradient, such as spectral projected gradient 2 (SPG 2). In the solving process of the SPG 2, the gradient information is obtained by calling the corresponding adjoint model. Therefore, the gradient-based algorithm can not solve a similar problem in the numerical models which do not have an adjoint model (such as CESM). We adopt the swarm intelligent algorithm based on the dimensionality reduction strategy to implement the CNOP method. The algorithm is suitable for solving such the nonlinear initial value problem in any numerical model. Besides, the algorithm has been optimized with a parallel framework to enhance performance. The major difference between related works (Jiang et al. and Dai et al.) and our works are summarized as follows: Model: T21L3 (with adjoint model) / CESM (has no adjoint model) Quantified variable: Geopotential height / SLP Perturbation variable(s): Potential vorticity / Zonal wind, meridional wind, temperature, specific humidity, surface pressure and surface geopotential Algorithm thought: gradient projection / swarm intelligent search The resolution and space structure (layer number, grid points) are also different in our work and related works.

3. The component set we selected for this case is F_2000 (F for short), with CAM, CLM, CICE and DOCN components. By default, the CAM component is stand-alone. As for the frequency, we set the parameter 'nhtfrq' to -24, which denotes the daily average. The variables of model input are listed in Table 2. They are zonal wind, meridional wind, temperature, specific humidity, surface pressure and surface geopotential.

4. Page 5, line 1: Yes, the NAOI stands for the NAO index, and the abbreviation is mentioned in Page 1, line 22.

5. Page 5, line 10: m is the number of spatial points, and n is the length of the time series.

6. Page 6, line 8-18: The constraint given in the initial value problem is to ensure the reasonability of these variables and avoid the appearance of abnormal values. Our work and the research of identifying sensitive areas for tropical cyclones have the same variables. The feasibility of restraining these variables using the constraint defined in equation (4) has been proved in [1-3].

7. Page 7, line 7-8 & Page 11, line 9: We run the component set F of CESM for 10 model years (only for winter, DJF) without perturbations, and gather the output of SLP on a daily average. Then the centralization SLP is obtained by subtracting the mean value of these 10 years and weighting according to its latitude.

8. Page 8, line 25: We are very sorry for our incorrect writing, and it was corrected.

9. Page 9, line 6: The original sentence is not accurate enough, thank you for pointing it out. The function of the CAM is to simulate the variation of the atmosphere.

10. Page 9, line 27: The asynchronous stream is the main carrier of asynchronous parallelism, and each stream can execute an independent task. The replication operation of memory corresponds to the replication engine of the hardware, and the function call corresponds to the execution engine of the kernel function. Therefore, these two operations can be conducted concurrently. To express the meaning more clearly, the sentence is modified into 'The function execution and data replication are overlapped using asynchronous streams'.

11. Page 17, line 19-22: The incorrect reference was corrected. Thank you.

12. Page 18, line 1: Sorry for the limited vocabulary, and we use other words instead.

13. Page 18, line 2: This sentence means that when the number of CPU nodes increases, the frequent communication would make the runtime of the CESM increase. We would try our best to improve the expression of this paper.

14. Page 19, line 15: It was corrected, thank you.

15. Page 21, line 2: We have replaced 'observably' by 'significantly' according to your suggestion.

We are truly grateful for your thoughtful comments. Based on these comments, we have tried our best to revise our manuscript. We would like to express our great appreciation to you for your comments on our paper.

* * *
[Figure]

**Supplement:**

[revised manuscript text omitted]

evidence that CNOP is a useful method to investigate the onset of the NAO event. In their studies, the T21L3 quasigeostrophic global spectral model, which is a simple three-level model designed by Marshall and Molteni, is applied under ideal conditions (Marshall and Molteni, 1993). Due to the feature of the T21L3 model, they adopted geopotential height as the characterized variable and selected potential vorticity as the input variable. For solving CNOP, they all used spectral projected gradient 2 (SPG2) algorithm (Birgin et al., 2001). The SPG2 was designed to solve the minimum problem with restraints by determining the gradients of the cost function (Guo-Dong, 2009). Several similar approaches have been also adopted to calculate the CNOP, such as the sequential quadratic programming (SQP) algorithm (Barclay et al., 1998) and the limited memory Broyden-Fletcher-Goldfarb-Shanno (L-BFGS) algorithm (Liu and Nocedal, 1989). These traditional methods of numerical optimal have been widely applied in the studies of the CNOP in the early years (Duan and Mu, 2006; Wang et al., 2012; Bo et al., 2014). Since these algorithms rely on gradient information, the corresponding adjoint model needs to be called to obtain the gradients of the initial condition in the solving process.

However, the traditional adjoint-based algorithms are not feasible to solve CNOP in complicated operational models that do not have an adjoint available (Wang, 2010). In addition, the past research suggests that the adjoint-based method would fail with large initial disturbance or long prediction time due to the strong nonlinearity of the dynamical model. The local CNOPs would be produced by the adjoint-based method with high probability when the objective function has multiple extreme values. In recent years, the swarm intelligence algorithms are gradually put forward to the research of the CNOP (Zheng et al., 2012; Yang et al., 2017). These algorithms determine the search directions and obtain the extremum by updating the position of the particles. Since the search process of these algorithms does not need any gradient information, they can be extended to the implementation of the CNOP using numerical models without the adjoint model. It is also indicated that the swarm intelligence method still achieves global CNOP and has a shorter run time in the situation of larger initial perturbations, longer prediction times, multiple extrema values (Zheng et al., 2017) and discontinuous objective functions (Mu et al., 2015). Although the above algorithms are effective, it is very time-consuming to calculate CNOP in the original space. To enhance the performance of solving CNOP with complex numeric models, the researchers proposed intelligent algorithms based on feature extraction. The algorithms transform the problems in original input space with high dimensions into the problems in low dimension space. At present, the tentative application of intelligent algorithms based on feature extraction in solving CNOP yielded considerable achievements. The principal component analysis based genetic algorithm (PCAGA) (Zhang et al., 2017a), the Modified Artificial Bee Colony Algorithm (MABC) (Ren et al., 2016), the dynamic search Fireworks Algorithm with linearly decreased dimension number strategy (ld-dynFWA) (Mu et al., 2017a) and PCA based Flower Pollination (PCAFP) (Yuan et al., 2016) have been successfully adopted in tropical cyclone adaptive observations, El Niño-Southern Oscillation and double-gyre variation. The CNOPs obtained by these methods have similar patterns and larger fitness values in comparison to the adjoint method. It is illustrated that the PCA-based intelligent algorithm is appropriate for high dimensional numerical models, especially the models without the adjoint model.

The objective of this paper is to find the OPRs which produce the NAO anomaly pattern and explore the effect of the nonlinear process. We study the case using the Community Earth System Model (CESM), which is an ocean-atmosphere coupled model without an adjoint model. Thus, traditional algorithms like SPG2 are inappropriate for this case. In this paper,

we select the particle swarm optimization (PSO) and genetic algorithm (GA) hybrid algorithm (PSO-GA), which is an effective swarm algorithm that has been previously proposed (Chang et al., 2013; Nik et al., 2016; Kumar and Vidyarthi, 2016; Agarwal and Srivastava, 2018), to solve CNOP for the NAO events. The parallel adjoint-free algorithm called PGAPSO is combined PSO-GA with the principal component analysis (PCA) strategy and is optimized with multiple frameworks. The OPRs obtained by the proposed algorithm steadily produce the SLP anomaly mode and trigger the high NAOI. Compared to the PCA-based PSO (PPSO), the algorithm is improved to avoid falling into the local optimum and accelerates convergence. After parallelized with MPI and CUDA, the speed-up ratio of the intelligent solution system reaches $40\times$ compared with its serial version.

The structure of this paper is organized as follows: Section 2 describes the CESM, and section 3 presents the CNOP method, the PGAPSO algorithm and the parallelization technique. Experiments and results are displayed in section 4. This paper ends with a conclusion and future work in section 5.

**2 Community Earth System Model**

The CESM (Kay et al., 2015) is a new generation of fully coupled climate models developed in 2010. It has been widely used to simulate the carbon cycle (Lehner et al., 2015), ocean currents (Large and Caron, 2015), soil moisture (Swenson and Lawrence, 2012), precipitation (Hagos et al., 2016) and other climate phenomena. As shown in Figure 1, the CESM is composed of seven geophysical model components, respectively Atmospheric (Community Atmosphere Model, CAM), Sea- ice (CICE), Land (Community Land Model, CLM), River-runoff (River Transport Model, RTM), Ocean (Parallel Ocean Program, POP), Land-ice (CISM), Ocean-wave (XWAV). The CESM also has a Coupler (CPL) that coordinates the time evolution of geophysical models and delivers information between these components.

[Figure]

**Figure 1.** Main components of CESM.

The atmospherical component in CESM 1.2.2 is used to simulate the NAO in this work. CAM version 5.3, which is a global atmospherical general circulation model developed from the NCAR CCM3, is released as the atmosphere component of CESM 1.2. The CAM incorporates an interactive aerosol model where aerosols interact with the tropospheric chemistry. The component set we selected is *F_2000* that includes CAM, CLM and CICE(prescribed mode) activated with SST data mode,

and the CAM component is stand-alone. The parameter *nhtfrq* is set to -24, which denotes the daily average. We perform the experiments on a $0.9°\times1.25°$ horizontal grid with 26 levels in the vertical. The region we focus on, which is also the NAO mainly located at, is a two-dimensional domain consisting of $65\times105$ grids with the North Atlantic area between $20°N$ and $80°N$ and between $90°W$ and $40°E$.

**3 CNOP and PGAPSO**

**3.1 CNOP**

The CNOP is a natural extension of the linear singular vector into the nonlinear regime, and is proposed to study predictability problems of weather and climate in numerical models (Mu et al., 2009). OPR is a kind of initial perturbation that can trigger the largest uncertainty in prediction, and it can be solved by the CNOP method. Specifically, the objective function achieves the maximum under the constraint condition at prediction time by superimposing OPRs on the basic state. In this experiment, we choose a blocking indicator proposed by Liu (Liu, 1994) to quantify the extent of the NAO events. The NAOI is defined as the projection of the SLP field on the NAO anomaly pattern:

$$NAOI = \frac{\langle SLP_{NAO}, SLP_d \rangle}{\langle SLP_{NAO}, SLP_{NAO} \rangle} \tag{1}$$

where $SLP_d$ is obtained by subtracting the climatological mean from SLP output, and $\langle \rangle$ denotes the inner product operation of vectors. $SLP_{NAO}$ denotes the NAO anomaly pattern acquired by the empirical orthogonal function (EOF) analysis. The EOF is a widely used tool to decompose the spatial-temporal distribution features in geonomy (Baldwin and Dunkerton, 1999). The procedures of EOF are listed as follows:

- Process the SLP historical data into anomaly values by subtracting the mean climate state of 10-year SLP time series data, recorded as $X_{m\times n}$. m denotes the number of spatial points, and n denotes the length of the time series.

- Calculate the covariance matrix $C_{m\times m}$ via: $C_{m\times m} = \frac{1}{n}X \times X^T$.

- Solve the eigenvalues ($\lambda_{1,...,m}$) and eigenvectors ($V_{m\times m}$) of $C_{m\times m}$ with the constraint condition: $C_{m\times m} \times V_{m\times m} = V_{m\times m} \times \Lambda_{m\times m}$.

- The eigenvectors corresponding to $\lambda_k$ is the $k^{th}$ column of $V_{m\times m}$, that is, $EOF_k = V(:,k)$.

In general, the first mode decomposed by EOF is chosen as the NAO anomaly pattern, which is illustrated in Figure 2. The NAO spatial pattern is manifested as a typical meridional dipole mode, which consists of the Iceland low pressure along with the North Atlantic subtropical high. In Figure 2, it is a positive phase of the NAO, presenting the mode with the negative anomalies in high latitude and the positive anomalies in low latitude.

[Figure]

**Figure 2.** The first mode of the EOF in SLP anomaly field concentrated in the North Atlantic region between $90°W$ - $40°E$, $20°N$ - $80°N$.

The procedure for solving CNOP can be regarded as the following extrema problem:

$$J(u_0^*)_{NAO^+} = \max_{\|u_0\| \leq \sigma} J(u_0) = NAOI(NAO^+)_{CNOP} - NAOI_{refer} \tag{2}$$

$$J(u_0^*)_{NAO^-} = \min_{\|u_0\| \leq \sigma} J(u_0) = NAOI(NAO^-)_{CNOP} - NAOI_{refer} \tag{3}$$

where $u_0$ is the vector of physics variables listed in Table 1. $J(u_0)$ is the objective function defined by the difference between
5    NAOI triggered by perturbation $u_0$ in the final state and NAOI in the reference state. In formula (2) and (3), the perturbation $u_0^*(NAO^+)$ makes $J(u_0)$ achieve the maximum, whereas $u_0^*(NAO^-)$ makes $J(u_0)$ achieve the minimum. According to the definition of the OPR, $u_0^*(NAO^+)$ is the OPR of the $NAO^+$, and $u_0^*(NAO^-)$ is the OPR of the $NAO^-$.

**Table 1.** The related variables included in the perturbations.

| Variable name | Description | Units |
|:---:|:---:|:---:|
| U | Zonal wind | $m/s$ |
| V | Meridional wind | $m/s$ |
| T | Temperature | $K$ |
| Q | Specific humidity | $kg/kg$ |
| PS | Surface pressure | $Pa$ |
| PHIS | Surface geopotential | $m^2/s^2$ |

$\sigma$ denotes the constraint condition of the OPRs. The constraint condition we used in this paper is consulted from a similar study in the field of the atmosphere. In the study of the identification of the sensitive areas for tropical cyclone using CNOP,

the summation of kinetic energy is chosen as the objective function, available relative potential and surface potential energy in the verification areas $D$ (Zhang et al., 2017a):

$$J(u_0)_{Trop} = \frac{1}{D} \int_D \int_0^1 [u'^2 + v'^2 + \frac{C_p}{T_r}t'^2 + R_a T_r (\frac{\pi'}{\pi_r})^2]d\sigma dD \tag{4}$$

where $J(u_0)_{Trop}$ is the objective function in the research of tropical cyclone. $u'$, $v'$, $t'$ and $\pi'$ are the initial perturbations of zonal wind, meridional wind, temperature and surface geopotential respectively. $C_p$ is the specific heat at the constant pressure which is set to 1005.7 $J \cdot kg^{-1}K^{-1}$ and $T_r$ is the reference temperature with a value of $270K$. $R_a$ denotes the ideal gas constant, and its value is set to 287.05 $J \cdot kg^{-1}K^{-1}$. $\pi_r$ is the reference static pressure with a value of 1000 $hPa$. In order to ensure the perturbations within a reasonable range, the constraint is set to 10% of the dry energy norm in the basic state, that is:

$$\sigma = 10\% * \frac{1}{D} \int_D \int_0^1 [U_0{}^2 + V_0{}^2 + \frac{C_p}{T_r}T_0{}^2 + R_a T_r (\frac{\Pi}{\pi_r})^2]d\sigma dD \tag{5}$$

We adopt the above constraint $\sigma$ since our work and the research of tropical cyclone have the same variables. The constraint is to ensure the reasonability of these variables and avoid the appearance of abnormal values. The feasibility of restraining these variables using this constraint has been proved in (Zhou and Mu, 2011; Zhang et al., 2017a, 2018). Combining the formula (1), (2) and (3), the objective function is described as follows:

$$\begin{aligned} J(u_0) &= \Delta NAOI \\ &= \frac{\langle M_{t_0 \to T}(U_0 + u_0) - M_{t_0 \to T}(U_0), SLP_{NAO} \rangle}{\langle SLP_{NAO}, SLP_{NAO} \rangle} \end{aligned} \tag{6}$$

where $M_{t_0 \to T}$ represents the nonlinear propagator that "propagates" the initial state in time $t_0$ to the prediction time $T$, and $U_0$ denotes the initial basic state. Therefore, $M_{t_0 \to T}(U_0)$ denotes the reference state at prediction time $T$. The objective function is the projection of the SLP field difference between the final state and the reference state on the NAO anomaly pattern.

**3.2 PGAPSO**

Under the resolution of *f09_g16* with an approximate grid spacing of $0.9° \times 1.25°$, the total dimensions of variables involved in the objective function are 5861376. It is difficult for the algorithm to solve the optimization problem in such high dimensions. Thus, we need to extract the feature of samples to reduce the data scale.

PCA is a traditional method for feature extraction and has been widely used in signal separation (Kasban et al., 2016), environment forecasting (ULSAUFIE et al., 2013) and pattern classification (Li et al., 2017), etc. In this paper, we adopt PCA to implement dimension reduction for sample data. After running 10-year integration (only in winter) on a daily average using

CESM, we extract the variables displayed in Table 1 from the model files. The size of $U$, $V$, $T$ and $Q$ is 26 (layer) $\times$ 192 (latitude) $\times$ 288 (longitude), and the size of $PS$ and $PHIS$ is 192 $\times$ 288. Each piece of sample is handled into a vector with one dimension (1 $\times$ 5861376), and the original sample is a matrix of 900 $\times$ 5861376 (containing 900 days). Then subtract the climatological mean of the 10-year data from each sample, and the obtained sample is weighted according to the area of the grid:

$$S_i = (S_i - \frac{1}{n} \sum S_i) * \cos(lat(i)) \ (i = 1, 2, \ldots, n) \tag{7}$$

where $S_i$ denotes the $i^{th}$ sample data and $n$ is the number of the samples. $lat(i)$ is the latitude of the $i^{th}$ row in the grid, and the weight is calculated approximately via the cosine value of the grid's latitude. Then the eigenvalues $(\lambda_1, \ldots, \lambda_n)$ and eigenvectors of the covariance matrix $SS^T$ are calculated to obtain principal components:

$$SS^T L = L\Sigma \tag{8}$$

where $L$ is the eigenvector matrix, and $\Sigma$ is a diagonal matrix whose entries in the main diagonal are the corresponding eigenvalues. The top $m$ columns of the eigenvectors $L$ sorting by their eigenvalues are selected as the principal components. The value of $m$ is determined by the contribution rate, which is defined as:

$$r = \frac{\sum_{i=1}^{m} \lambda_i}{\sum_{i=1}^{n} \lambda_i} \tag{9}$$

In this work, $m$ is set to the minimum number of columns that meet the contribution rate of 95%. The reduced space with $m$ dimensions is far smaller than the original one.

To obtain the extremum of the objective function, we adopt a hybrid algorithm improved from two classical algorithms, PSO and GA. The PSO is a type of intelligent heuristic algorithm to solve the problem with NP property (Kennedy, 2011). The position with the best fitness value is searched by tracing individual optimal positions and the optimal global position in the meantime. The flow of the algorithm is described in brief: (1) Initialize the speed ($V$) and position ($X$) of particle swarm with random values. The random values obey the normal distribution and ensure the perturbations satisfy the constraints. (2) For each particle $i$, the position vectors in reduced space need to be restored into original space via $X_i' = X_i \cdot L_{1,\ldots,m}$, thereinto, $L$ is an m times m eigenvector matrix. Then superpose the perturbation $X_i'$ on the basic state. When the model integration is finished, calculate the fitness value of each particle through the formula (6) and record its optimal position ($X_{pb}$) along with the global optimal position ($X_{gb}$). (3) Update the position and speed of each particle. The updating formula is as follows:

$$\begin{cases} V_i^{k+1} = \omega_k V_i^k + c_1 r_1 (X_{pb}^k - X_i^k) + c_2 r_2 (X_{gb}^k - X_i^k) \\ X_i^{k+1} = X_i^k + V_i^{k+1} \end{cases} \tag{10}$$

where $V_i^k$ is the speed of particle $i$ for step $k$ and $V_i^{k+1}$ is for step $k+1$. $c_1$ is the self-awareness coefficient for the historical self-optimal position and $c_2$ is the social-awareness coefficient for global optimal position of all particles. The empirical value of $c_1$ and $c_2$ are both set to 2. $r_1$ and $r_2$ are random float numbers with uniform distribution in $[0,1]$. $X_{pb}^k$ refers to the position of particle $i$ where objective function acquires the maximum(minimum) in $k$ steps, and $X_{gb}^k$ represents the position where objective function achieves global extrema in $k$ turns. Both position vectors and speed vectors are in reduced space with $m$ dimensions. $\omega_k$ is the weight parameter and calculated by:

$$\omega_k = \omega_{max} - \frac{\omega_{max} - \omega_{min}}{iter_{max}} * iter \tag{11}$$

where $iter$ is the current number of step, and $iter_{max}$ is set to 100.

In PGAPSO, PSO is viewed as the main body of the search process, and the GA further optimizes the position. As a meta-heuristic algorithm, the GA derives from natural selection (Goldberg and Holland, 1988). When the fitness value is obtained in step (2) of PSO, the particles are fed into GA for further search. A portion of particles in the existing population are selected according to their fitness value to breed a new generation. The selection operation is performed on roulette strategy, that is to say, the probability that each individual is selected is equal to the ratio of its fitness value to the total fitness value of the entire population:

$$p_s = \frac{J(u_{X_i'})}{\Sigma J(u_{X'})} \tag{12}$$

After that, the selected parents generate new individuals via crossover:

$$X_a'\{x_s,\ldots,x_e\} = X_b\{x_s,\ldots,x_e\}$$
$$X_b'\{x_s,\ldots,x_e\} = X_a\{x_s,\ldots,x_e\} \tag{13}$$

Then the new generation mutates with probability $p_m$ in a single position to avoid genetic drift. The fitness value of each new generation is compared against its parents, and the best position is recorded. If the new individual has a better fitness value compared with the global best position, the global best position ($X_{gb}^k$) would be replaced by the new position. With the optimal local position and global optimal position, the speed and position of particles are updated using formula (10). The final global fitness value is obtained until the $iter$ reaches $iter_{max}$ or the norm of particles' speed reaches the specific threshold.

**3.3 Parallelization**

The computation of CNOP in CESM is quite time-consuming. With 48 CPU cores, 30 particles and 100 iterations, it takes about 13.75 days to obtain the OPRs in the serial program. For PGAPSO to operate more effectively, multiple parallel techniques and frameworks are adopted in this work.

**3.3.1 CESM Parallelization**

The role of the CAM component in CESM is to simulate the variation of atmosphere, and the largest variation can be discovered by objective function using PGAPSO. With high resolution, the input data handled for integration in nonlinear processes of CAM possess the features as massive variables, high dimensions and complexity, which makes the invocation for CAM become the primary time-consuming task in the whole program. Although CESM has already been parallelized using Message Passing Interface (MPI) and Open Multi-Processing (OpenMP), it is still time-consuming.

Recently, the Graphics Processing Unit (GPU) has been widely used in accelerating numerical models. Since GPU is suitable for parallel computing on a large scale, it can significantly improve the execution performance of climate models. A parallel scheme for Community Climate System Model (CCSM) has been proposed to shorten the runtime of climate prediction by porting the radiation module onto GPUs (Coleman and Feldman, 2013). The module was parallelized using the inline method and communicated with MPI routines. A cloud analysis scheme called Goddard Cumulus Ensemble (GCE) in Weather Research and Forecasting (WRF) was highly expedited using NVIDIA Tesla K40 with 2880 cores (Huang et al., 2015). Compared to the CPU-based parallel version running on 4 nodes, the GPU-based scheme performed faster. As for CESM, the novel asynchronous execution strategy has provided significant performance benefits (Korwar et al., 2013). The most time-consuming routines have been accelerated via OpenACC directives and achieved a speedup of $1.19\times$-$1.53\times$ for the entire model. Another attempt for accelerating CESM was to port CESM along with a rewritten vertical remapping scheme onto GPUs (Carpenter et al., 2013). The results indicated that the performance of the optimized subroutine was improved substantially. Related works show that GPU is an alternative approach to enhance the performance of the climate model.

In this work, we port several time-consuming subroutines in CAM onto GPUs through the PGI CUDA Fortran interface. After analysis run time using *pref*, shown in Figure 3, subroutine *radclwmx* and *radabs* both consume longer runtime compared with other subroutines. These two subroutines are both optimized with the CUDA platform. Simultaneously, kernel directives and OpenACC directives are used to implement the simplification of specific operations on the device. The function execution and data replication are overlapped using asynchronous streams. 
[revised manuscript text omitted]

The trends for these three states at another start date are shown in Figure 6. This figure also illustrates for 5-day optimization, 7-day optimization and 15-day optimization. In Figure 6, similar to Figure 5, the NAOI triggered by CNOP always has a big gap with the reference flow and achieves an abnormal high value in the final period. Similarly, the flows of $NAO^+$ and $NAO^-$ both noticeably deviate from the reference flow in the last few days.

[Figure]

**Figure 6.** Same as Figure 5, but for another start date.

To evaluate the NAOI of CNOPs more visually, Table 3 reports the incremental values of the NAOI with different simulation time in Figure 5 and Figure 6. From Table 3, the difference between the NAOI in the final state and the NAOI in reference state increases when the integration time becomes longer. We can also find that the result depends on the start date. Although large discrepancy exists between the $\Delta NAOI$ with a simulation time of 15 days in Figure 5 and Figure 6, the algorithm can always find the CNOPs that can cause the abnormal state, and $|\Delta NAOI|$ is far greater than 1.

**Table 3.** The increment value of NAOI with different simulation time in Figure 5 and Figure 6.

[revised manuscript text omitted]

5 ### 4.3 Performance Analysis

In order to demonstrate the performance improvement of parallel PGAPSO adopted in this paper, Figure 10 compares the runtime of parallel PGAPSO and serial PGAPSO for one iteration. The runtime of CESM is the performance bottleneck of the algorithm, which can be broken by running in parallel. Our parallel scheme using MPI implements the simultaneous execution of multiple particles to solve the problem. From Figure 10, we can see that when the number of CPU cores is more than 840, it
10 will take longer to run the serial algorithm. CESM has been paralleled with MPI and OpenMP; when the number of CPU cores increases to the critical point, the frequent communication would make the runtime of the CESM increase. The speedup ratio of parallel PGAPSO compared with serial PGAPSO is displayed in Table 4. The speedup ratio increases with the rise of the CPU cores' number. With assigning CPU cores to multiple tasks, the execution time of parallel PGAPSO continues to decline, while the serial PGAPSO takes longer owing to communication. With 1080 CPU cores, PGAPSO based on the parallel scheme
15 achieves a speedup of 40× compared to its serial version.

**Table 4.** The speedup of parallel PGAPSO compared with serial PGAPSO.

[revised manuscript text omitted]

Zhang, X., Mu, M., Wang, Q., and Pierini, S.: Optimal precursors triggering the Kuroshio Extension state transition obtained by the Condi-

20  tional Nonlinear Optimal Perturbation approach, Advances in Atmospheric Sciences, 34, 685–699, 2017b.

Zheng, Q., Dai, Y., Zhang, L., Sha, J., and Xiaoqing, L. U.: On the Application of a Genetic Algorithm to the Predictability Problems Involving "On-Off" Switches, Advances in Atmospheric Sciences, 29, 422–434, 2012.

Zheng, Q., Yang, Z., Sha, J., and Yan, J.: Conditional nonlinear optimal perturbations based on the particle swarm optimization and their applications to the predictability problems, Nonlinear Processes in Geophysics, 24, 101–112, 2017.

25  Zhou, F. and Mu, M. U.: The Impact of Verification Area Design on Tropical Cyclone Targeted Observations Based on the CNOP Method, Advances in Atmospheric Sciences, 28, 997, 2011.